# Congenital Zika syndrome: A systematic review

**Danielle A. Freitas**[1☯¤a]*, **Reinaldo Souza-Santos**[1☯]*, **Liege M. A. Carvalho**[2‡], **Wagner B. Barros**[2‡], **Luiza M. Neves**[3‡], **Patrícia Brasil**[2‡], **Mayumi D. Wakimoto**[2☯¤b]*

**1** National School of Public Health, Oswaldo Cruz Foundation, Rio de Janeiro, RJ, Brazil, **2** Evandro Chagas National Institute of Infectious Diseases, Oswaldo Cruz Foundation, Rio de Janeiro, RJ, Brazil, **3** Fernandes Figueira Institute, Oswaldo Cruz Foundation, Rio de Janeiro, RJ, Brazil

☯ These authors contributed equally to this work.
¤a Current address: National School of Public Health, Oswaldo Cruz Foundation, Rio de Janeiro, RJ, Brazil
¤b Current address: Evandro Chagas National Institute of Infectious Diseases, Oswaldo Cruz Foundation, Rio de Janeiro, RJ, Brazil
‡ These authors also contributed equally to this work.
* dafufrj@gmail.com (DAF); mayumidw@gmail.com (MDW); rssantos@ensp.fiocruz.br (RSS)

## Abstract

### Background

The signs and symptoms of Zika virus infection are usually mild and self-limited. However, the disease has been linked to neurological complications such as Guillain-Barré syndrome and peripheral nerve involvement, and also to abortion and fetal deaths due to vertical transmission, resulting in various congenital malformations in newborns, including microcephaly. This review aimed to describe the o signs and symptoms that characterize the congenital Zika syndrome.

### Methods and findings

A systematic review was performed with a protocol and described according to the recommendations of the Preferred Reporting Items for Systematic Reviews and Meta-Analyses statement. The search strategy yielded 2,048 studies. After the exclusion of duplicates and application of inclusion criteria, 46 studies were included. The main signs and symptoms associated with the congenital Zika syndrome were microcephaly, parenchymal or cerebellar calcifications, ventriculomegaly, central nervous system hypoplasia or atrophy, arthrogryposis, ocular findings in the posterior and anterior segments, abnormal visual function and low birthweight for gestational age.

### Conclusions

Zika virus infection during pregnancy can cause a series of changes in the growth and development of children, while impacting the healthcare system due to the severity of cases. Our findings outline the disease profile in newborns and infants and may contribute to the development and updating of more specific clinical protocols.

**Data Availability Statement:** All relevant data are within the manuscript and its Supporting Information files.

**Funding:** The author(s) received no specific funding for this work.

**Competing interests:** The authors have declared that no competing interests exist.

**Abbreviations:** ZIKV, Zika virus; CNS, central nervous system; SINASC, System on Live Births; WHO, World Health Organization; PICO, population, intervention, comparison, outcome; MINORS, Methodological Index for Non-Randomized Studies; JBI, Joanna Briggs Institute) critical appraisal checklist for case reports; TORCH, Toxoplasmosis, Rubella, Cytomegalovirus and Herpes; CMV, Cytomegalovirus.

## Introduction

Zika virus (ZIKV) is a flavivirus of the family Flaviviridae, isolated initially in non-human primates in Uganda (1947), and in humans (1954) in Nigeria, Africa [1–4].

The first recorded outbreak was on the Yap Islands of Micronesia in 2007, followed by an epidemic in French Polynesia in 2013 and 2014 [5]. In Brazil, the first cases of the disease were reported in May 2015 [6].

Compared to other arboviruses such as dengue and chikungunya, ZIKV infection involves additional transmission routes. Besides transmission by *Aedes* mosquitoes, the risk of ZIKV spread may even be greater because the virus can also be transmitted via sexual relations, blood transfusions, and vertical transmission, and ZIKV has also been detected in urine and saliva [7–11].

The laboratory diagnosis of ZIKV infection is limited by the high cost and cross-reaction with other flaviviruses [12,13], and then protocols for clinical diagnosis, in the context of simultaneous infection by other arboviruses, need to be implemented to define cases of ZIKV infection among pregnant women who have a rash [14].

Most individuals infected with the Zika virus either do not develop symptoms or have mild and self-limited signs [15–17]. However, the disease has been linked to several neurologic manifestations in children and adults such as Guillain-Barré syndrome and peripheral nerve involvement, and ophthalmic complications such as retinal and optic nerve abnormalities. Vertical transmission has been associated with spontaneous abortion and stillbirth, and also with congenital malformations in newborns including but not limited to microcephaly [5,15,18–33].

According to the Brazilian Information System on Live Births (SINASC), the prevalence of microcephaly from 2000 to 2014, prior to ZIKV circulation, was 5.5 per 100,000 live births. However, in 2015, with the introduction of the virus and the onset of the first cases of microcephaly, the prevalence rate increased to 54.6 per 100,000 live births, i.e., an increase of 9.8 times [34].

The emergency committee of the World Health Organization (WHO) declared Zika a Public Health Emergency of International Concern in February 2016. The emergence of a disease with potentially severe impact on pregnant women and newborns triggered the search for global partnerships and the joint efforts of governments and experts to describe the infection's pathophysiology and deal with the related clinical and social challenges [35,36].

The association between ZIKV infection and cases of microcephaly was first reported in 2015 in Brazil [28]. Detection of ZIKA infection during pregnancy has been found to be harmful to the fetus and can lead to fetal death and other abnormalities in newborns [15,19,37].

Microcephaly is only one of the possible complications found in neonates exposed to ZIKV during pregnancy and comprising the congenital Zika syndrome (CZS) [28,38–41]. There is sufficient evidence to support a causal link between ZIKV and congenital anomalies, at both the population and individual levels [38,42–49].

The systematic reviews published to date have addressed causality, transplacental transmission of ZIKV, the effect of sexual transmission, and antibody-dependent enhancement of viral teratogenicity, a different approach from that of the present review [38,42–49]. This review aimed to determine the signs and symptoms that characterize the congenital Zika syndrome and contribute to a more accurate and timely diagnosis.

## Methods

### Literature search

This review was performed with a protocol and described according to the recommendations of the Preferred Reporting Items for Systematic Reviews and Meta-Analyses statement [50]

and is registrered in PROSPERO (CRD42020151754) in 27 October 2019. We used PICO (population, intervention, comparison, outcome) as a search strategy tool as described in S1 Table.

A systematic search was conducted in Medical Literature Analysis and Retrieval System Online (MEDLINE), Latin American and Caribbean Health Sciences Literature, Scientific Electronic Library Online, Web of Science, Excerpta Medica Database (EMBASE), Scopus databases, and Biblioteca Virtual em Saúde (BVS) to identify studies assessing signs and symptoms associated with congenital Zika virus syndrome. Additionally, manual search was performed for bibliographic references of the selected articles and grey literature databases were also included to minimize publication bias [51–55].

The search descriptors used for MEDLINE were as follows: "Zika or zikv" and "pregnan*" or "children or newborn or infant" and "congenital or congenital abnormalities" or "birth defects or malformations or microcephaly". The search strategy was adapted according to the characteristics of each database. The complete search strategies used are presented in S3 Appendix.

There were no language restrictions in the database searches.

## Selection

Article selection was performed by two authors independently (DAF and WB), and disagreements were resolved by discussion and consensus. Studies were included if they reported data describing signs and symptoms of fetal growth impairment and altered development in children exposed to Zika virus during pregnancy, among laboratory-confirmed mothers and/or children < = 2 years old. First, the titles and abstracts retrieved by the search were read and excluded as follows: editorials, letters, guidelines, and reviews. Secondly, studies potentially eligible for inclusion were read in full text and the same inclusion and exclusion criteria were applied.

## Data extraction and quality assessment

Data were extracted by two authors independently (DAF and WB), and reviewed by the other (MDW). Discrepancies were resolved by discussion and consensus reached between the authors. A standardized data extraction form was made for the review. This form, available upon request, included the following sections: identification of the study (authors, journal and year of publication, language); study characteristics (design, total number of patients, period); study population (age, gestational age at infection, diagnostic method, differential diagnosis, pregnancy outcomes, birthweight, and head circumference); and analytical method. Assessment of the methodological quality of the observational studies was based on the Methodological Index for Non-Randomized Studies (MINORS) [56]. The instrument consists of 12 items, the first eight being specific for noncomparative studies. To assess the quality of case reports, we used JBI (Joanna Briggs Institute) critical appraisal checklist for case reports [57]. Each article was evaluated by two authors (MDW and DAF) independently, and disagreements were resolved by consensus.

## Data synthesis and analysis

The studies were described according to country, study and sample characteristics, clinical examination, imaging tests, autopsy findings, complementary tests (laboratory, radiography, electrocardiography), and placental alterations.

## Results

The search strategy yielded 2,048 studies. After exclusion of duplicates and application of inclusion criteria to the titles, abstracts, and full text, 81 studies were eligible for full text reading. Based on the full text reading, 46 articles were included in this review (Fig 1).

Among the articles, more than 90% presented end points appropriate to their aim and consecutive inclusion of patients according to Minors. Less than 20% of the studies reported adequately on the following criteria: "loss to follow-up less than 5%", "prospective calculation of

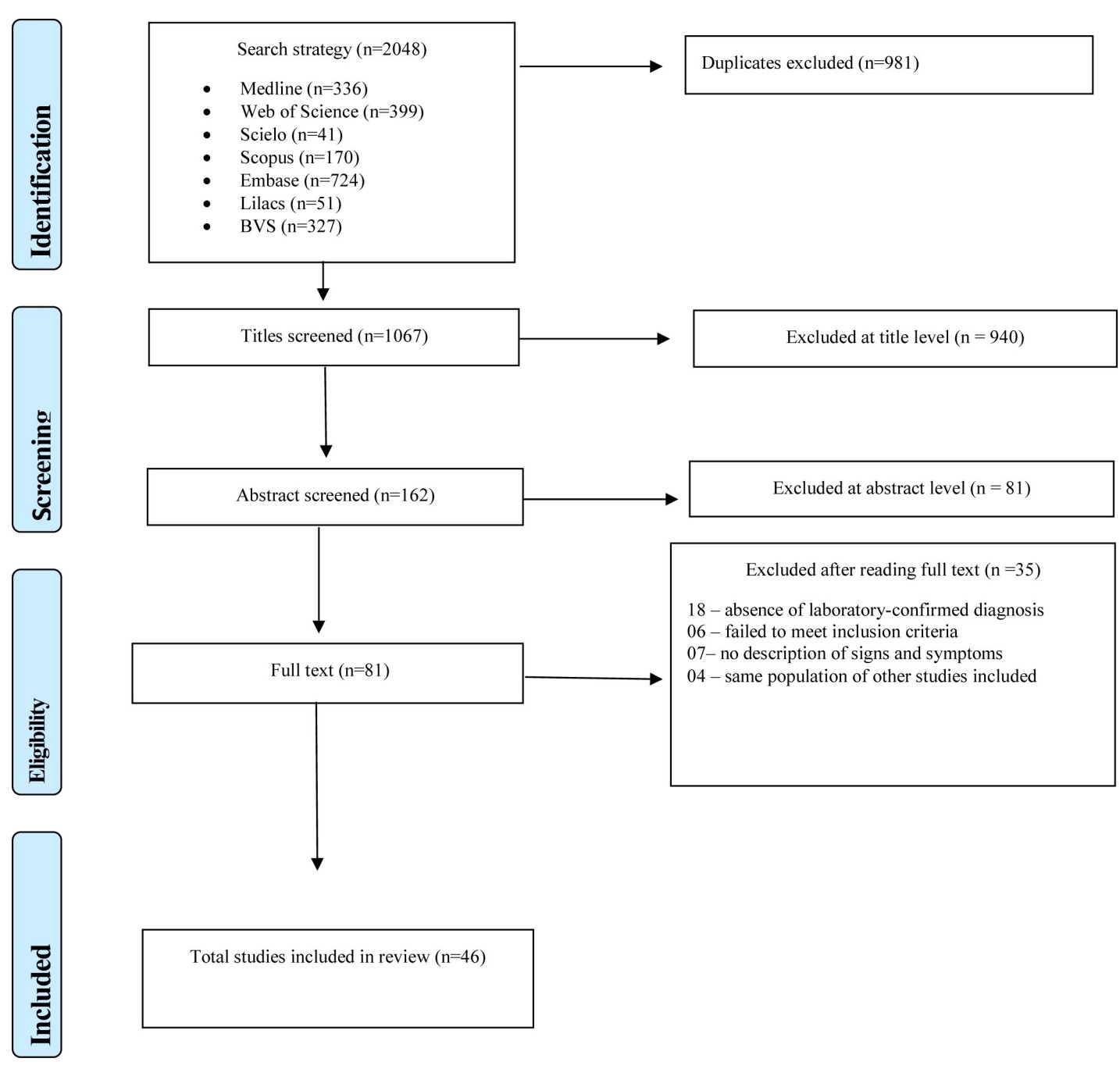

**Fig 1. Flowchart of review process.**

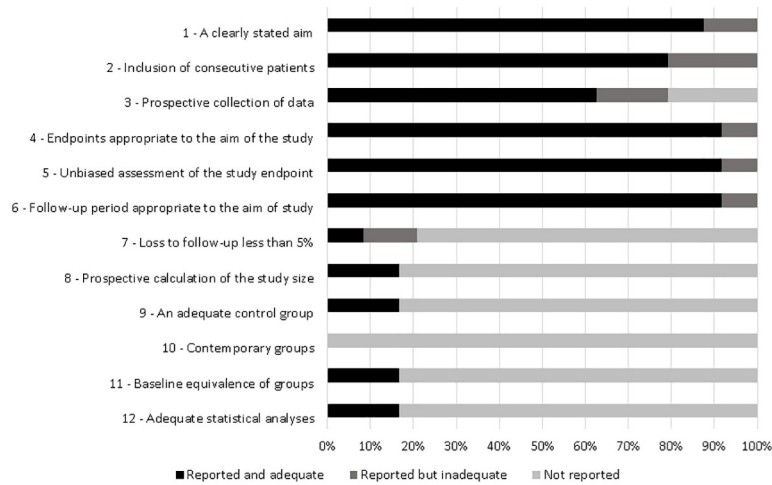

**Fig 2. Quality assessment of the studies included in the systematic review based on Methodological Index for Non-Randomized Studies (MINORS).**

the study size", "an adequate control group", "Contemporary groups", "baseline equivalence of groups" and "adequate statistical analyses" (Fig 2). From all case reports 36% did not describe patient´s history adequately. The other items were adequate in more than 90% of studies and two were not applicable (Fig 3).

Most of the studies were performed in Brazil (59%), the epicenter of reported cases. Regarding study design, most of the studies were case reports (n = 22) or case series (n = 20), and there were two cohort studies, one case-control, and one cross-sectional study. All the studies reported at least one laboratory method for the diagnosis of women and/or children exposed to Zika virus during pregnancy. Most of the studies described signs and symptoms in children exposed to Zika virus in the first and second trimesters (Table 1).

Regarding study design, most of the studies were case reports (n = 22) or case series (n = 20), and there were two cohort studies, one case-control, and one cross-sectional study. All the studies reported at least one laboratory method for the diagnosis of women and/or children exposed to Zika virus during pregnancy. Most of the studies described signs and symptoms in children exposed to Zika virus in the first and second trimesters (Table 1).

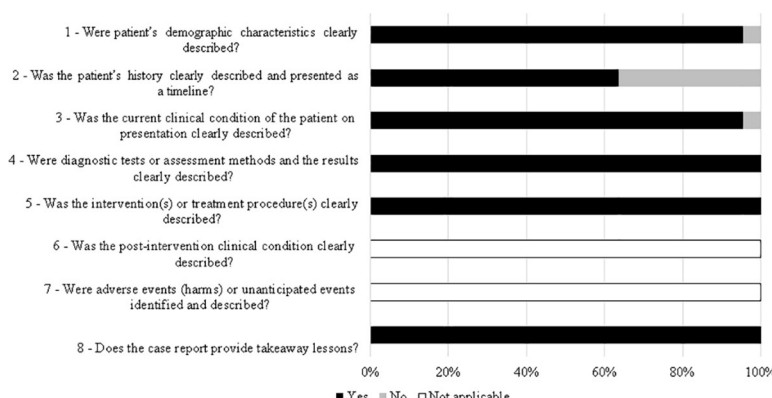

**Fig 3. Quality assessment of the studies included in the systematic review based on JBI (Joanna Briggs Institute) critical appraisal checklist for case reports.**

**Table 1. Characteristics of the studies included.**

| Author | Country | Type of study | Sample size | Diagnostic method (ZIKV) | Torch | Mother's age (average) | Gestational age of infection (average) | Gestational age of outcome of pregnancy (average) | Outcome of pregnancy | Weight Grams (mean) | Cephalic perimeter (Centimetermean) | Sex ratio M/F |
|---|---|---|---|---|---|---|---|---|---|---|---|---|
| Pomar et al (2017) [58] | French Guiana | Cohort | 700 | RT-PCR*¥€, IgM*#, IgG*# | Yes | 28 | 1st trimester: 80, 2nd trimester:96, 3rd trimester:125 | - | Therapeutic abortion:2, Fetal death 9:, Live birth:125, Ongoing pregnancy: 165 | - | - | - |
| Honein et al (2017) [59] | EUA | Case series | 442 | RT-PCR*#¶, IgM*#, PRNT€¥ | No | - | | - | Fetal death:5, Live birth:21 | - | - | - |
| Brasil et al (2016) [19] | Brazil | Cohort | 207 | RT-PCR* | Yes | 31 | 1st trimester:16, 2nd trimester:27, 3rd trimester:24 | 37.9 | Miscarriage:7, Fetal death:2, Live birth:49 | - | - | - |
| de Araújo et al (2016) [23] | Brazil | Case-control | 94 | RT-PCR*#¶, IgM*#, PRNT* | Yes | 24 | | 36.37 | Fetal death:1, Live birth:31, †3 | 2077 | - | 0.74 |
| Zin et al (2017) [60] | Brazil | Case series | 112 | RT-PCR*€¥ | Yes | - | 1st trimester:14, 2nd trimester: 8, 3rd trimester:2 | - | Live birth:24 | - | - | - |
| Aragão et al (2017) [61] | Brazil | Case series | 12 | IgM# | Yes | - | 1st trimester: 6, 2nd trimester:3, Without symptoms:3 | 38.16 | Live birth:12 | - | 29.04 | 1.4 |
| Ventura et al (2017) [62] | Brazil | Cross-sectional | 32 | IgM¶ | Yes | 26 | 1st trimester:13, 2nd trimester:9, 3rd trimester 3, Without symptoms:1 | 37.9 | Live birth:32 | 2627 | 28.7 | 1.28 |
| Schaub et al (2017) [63] | Martinique | Case series | 8 | RT-PCR*¥ | Yes | 26 | 1st trimester:6, Without symptoms: 2 | 28.12 | Therapeutic abortion:7, Live birth:1 | 1218 | 22.69 | 1 |
| Carvalho et al (2016) [64] | Brazil | Case series | 19 | RT-PCR, IgM | Yes | 26 | 1st trimester:11, 2nd trimester:2, Without symptoms:6 | 38.83 | Fetal death:1, Live birth:18 | 2554,6 | - | 0.9 |
| Aragão et al (2017) [65] | Brazil | Case series | 23 | IgM*#, IgG*# | Yes | - | 1st trimester:, 2nd trimester:, 3rd trimester, Without symptoms: | 37.43 | Live birth:23 | - | 28.91 | 1.3 |
| Hazin et al (2016) [66] | Brazil | Case series | 23 | IgM# | Yes | - | - | - | Live birth:23 | - | - | 1.3 |
| Oliveira-Szejnfeld et al (2016) [67] | Brazil | Case series | 45 | RT-PCR*�◊¥ | Yes | - | - | 39.1 | Fetal death:3, Live birth:14 | - | 28.93 | - |
| Besnard et al (2016) [68] | French Polynesian | Case series | 19 | RT-PCR¥ | Yes | - | - | - | Therapeutic abortion:11, Live birth:8 | - | - | - |
| van der Linden et al (2016) [69] | Brazil | Case series | 7 | IgM¶ | Yes | - | 1st trimester:4, Without symptoms:3 | - | Live birth:7 | - | 28.85 | - |
| Melo et al (2016) [70] | Brazil | Case series | 11 | RT-PCR¥◊, PRNT# | Yes | - | 1st trimester:9, 2nd trimester:1, Without symptoms:1 | 39.09 | Therapeutic abortion1:, Fetal death:2, Live birth:8 | - | 24.36 | - |
| Van der Linden et al (2017) [71] | Brazil | Case series | 13 | IgM#¶ | Yes | - | 1st trimester:4, 2nd trimester:2, Without symptoms:8 | 38.3 | Live birth:13 | 2971 | 32.23 | 2.25 |
| Sanín-Blair et al (2017) [72] | Colombia | Case report | 3 | RT-PCR¥ | No | 23 | 1st trimester: 3 | - | Ongoing pregnancy: 3 | - | - | - |
| Meneses et al (2017) [73] | Brazil | Case series | 87 | PRNT# | Yes | 24 | 1st trimester:47, 2nd trimester:17, 3rd trimester:2 | 38.5 | Live birth:87†3 | 2575 | 28.1 | - |
| Fernandez et al (2017) [74] | EUA | Case report | 4 | RT-PCR#◊ | Yes | - | | 25.25 | Therapeutic abortion:4 | - | - | 1 |
| Martines et al (2016) [40] | Brazil | Case report | 5 | RT-PCR◊¶, Immunohistochemistry◊ | Yes,1 no | 26 | 1st trimester: 5 | 27.4 | Miscarriage:2, Live birth:3, †2, ‡1 | 2399 | 29.66 | 0.5 |
| Sousa et al (2017) [75] | Brazil | Case series | 7 | RT-PCR◊¶ | No | - | 1st trimester: 7 | 36 | Live birth:7† | - | - | 1.33 |
| Guillemette-Artur et al (2016) [76] | French Polynesia | Case report | 3 | RT-PCR¥, IgM# | Yes | - | 1st trimester:2 | 29.2 | Therapeutic abortion:4 | - | - | 0.5 |
| Castro et al (2017) [77] | Brazil | Case series | 8 | RT-PCR¥ | Yes | 29 | 1st trimester: 6 | 38.27 | Live birth:8 | - | 25.17 | - |
| Ventura et al (2016) [78] | Brazil | Case series | 40 | IgM¶ | Yes | 26 | 1st trimester: 13, 2nd trimester: 11, 3rd trimester:3 | Pre-term 11 Term 28 Post-term 1 | Live birth:24 | 2674 | 29.5 | 1.1 |

**Table 1.** (Continued)

| Author | Country | Type of study | Sample size | Diagnostic method (ZIKV) | Torch | Mother's age (average) | Gestational age of infection (average) | Gestational age of outcome of pregnancy (average) | Outcome of pregnancy | Weight Grams (mean) | Cephalic perimeter (Centimetermean) | Sex ratio M/F |
|---|---|---|---|---|---|---|---|---|---|---|---|---|
| Parra-Saavedra et al (2017) [79] | Colombia | Case series | 17 | RT-PCR*€¥ | No | 24 | 1st trimester: 10, 2nd trimester:4, Without symptoms:3 | 31 | Live birth:17 | 2198 | 27.14 | 0 |
| Del Campo et al (2017) [80] | Brazil | Case series | 83 | IgM#¶ | Yes | - | - | - | Live birth:12 | - | - | - |
| Chimelli et al (2017) [81] | Brazil | Case series | 10 | RT-PCR*◊, Immunohistochemistry◊ | Yes | - | 1st trimester:6, 2nd trimester: 3, 3rd trimester1 | 37.5 | Fetal death:3, Live birth:7 | - | 31.3 | 1.5 |
| Schaub et al (2017) [82] | France | Case series | 14 | RT-PCR*¥¶€ | Yes | - | 1st trimester: 13, 2nd trimester:1 | 27.36 | Miscarriage:12, Fetal death:1, Live birth:1 | - | - | - |
| Mattar et al (2017) [83] | Colombia | Case report | 1 | RT-PCR€, IgM* | Yes | 33 | 2nd trimester: 1 | 39 | Live birth:1 | - | 27.5 | - |
| Culjat et al (2016) [84] | EUA | Case report | 1 | RT-PCR€, IgM*#, PRNT€ | Yes | 32 | 1st trimester: 1 | 39 | Live birth:1 | 2063 | 27.3 | 0 |
| Souza et al (2016) [85] | Brazil | Case report | 1 | RT-PCR¥, IgM*¶ | Yes | 17 | 2nd trimester: 1 | 39 | Live birth:1 | 2565 | 28 | - |
| Perez et al (2016) [86] | Spain | Case report | 1 | RT-PCR*ᵟ◊¥, IgM*, IgG*, PRNT* | Yes | 20 | 1st trimester:1 | 21 | Therapeutic abortion:1 | 295 | 17 | 0 |
| Van der Linden et al (2017) [87] | Brazil | Case report | 2 | IgM¶ | Yes | - | 1st trimester: 2 | 35.5 | Live birth:23 | 1925 | | 1 |
| Souza et al (2016) [88] | Brazil | Case report | 2 | RT-PCR*# | Yes | 27 | Without symptoms:2 | 38.5 | Live birth:23 | 2445 | 35.25 | 0 |
| Vorona et al (2016) [89] | EUA | Case report | 1 | RT-PCR*¥ | No | 50 | 1st trimester:1 | - | Therapeutic abortion:1 | - | - | - |
| Zacharias et al (2017) [90] | EUA | Case report | 1 | RT-PCR◊ᵟ€ | No | - | | 36 | Live birth:1† | 1885 | 28.5 | - |
| Werner et al (2016) [91] | Brazil | Case report | 1 | RT-PCR*#, IgM* | Yes | 31 | 1st trimester:1, 2nd trimester:, 3rd trimester, Without symptoms: | 37 | Live birth:1 | 2450 | 28 | - |
| Freitas et al (2016) [92] | Brazil | Case report | 1 | IgM# | No | - | | 38 | Live birth:1† | 1892 | - | - |
| Moron et al (2016) [93] | Brazil | Case report | 1 | IgG# | Yes | 31 | 2nd trimester: 1 | - | Live birth:1 | - | 30.5 | 0 |
| Frutos et al (2017) [94] | Argentina | Case report | 1 | RT-PCR*¥, IgM¶# | Yes | 27 | 1st trimester: 1 | - | Live birth:1 | 3390 | 31 | - |
| Acosta-Reyes et al (2017) [95] | Colombia | Case report | 2 | RT-PCR¥◊ | Yes | 19 | 1st trimester: 1, 2ndtrimester:1 | 23.5 | Therapeutic abortion:2 | - | - | - |
| Vesnaver et al (2016) [96] | Slovenia | Case report | 1 | RT-PCR#◊€, Immunohistochemistry◊ | Yes | 25 | 1st trimester: 1 | 32 | Fetal death:1 | 1470 | 26 | 1 |
| Sarno et al (2016) [97] | Brazil | Case report | 1 | RT-PCR◊, IgM* | Yes | 20 | Without symptoms:1 | 32 | Fetal death:1 | 930 | - | - |
| Driggers et al (2016) [45] | EUA | Case report | 1 | RT-PCR*#. IgM#*, IgG#* | Yes | 33 | 1st trimester: 1 | 21 | Therapeutic abortion:1 | - | - | - |
| Oliveira et al (2016) [98] | Brazil | Case report | 2 | RT-PCR¥ | No | - | - | - | Ongoing pregnancy: 2 | 1098 | 23.75 | - |
| Narero et al (2016) [99] | Panama | Case report | 1 | RT-PCR#, IgM* | Yes | 20 | - | 32 | Live birth:1† | 1020 | 23 | 1 |

Blood / serum / mother urine

*, Child blood / serum

#, Child tissues

◊Amniotic fluid

¥, Liquor

¶Placenta

€, Neonatal death

†, Post neonatal death

‡ Not reported -

Regarding clinical signs, microcephaly was described in 21 articles [19,23,40,59,60,62, 63,65,69,71,73,76,78,80,84,87,90,92,93,99,100], hypertonicity in eight [19,62,63,71,80,87,93,94], seizures in seven [19,40,62,70,71,73,84], and neurological crying/irritability in the early months of life in six [19,71,80,83,87,94] (Table 2). A frequently observed clinical sign was arthrogryposis, reported in 12 articles [40,61,69,71,73,75,80,85–87,90,92]. Other osteoskeletal signs included clubfoot [85,87] and hip dysplasia [19,71]. Morphological changes of the head were described in ten articles [19,40,65,69,71,73,80,83,84,87] and overriding sutures or closed fontanels in five [70,71,80,83,84].

The most frequent ophthalmic abnormalities occurred in the posterior segment, found in 22% of the reviewed articles which included lesions of the retina (focal pigmentary retina mottling, chorioretinal atrophy and/or coloboma) and optic nerve (pallor, atrophy, increased excavation, hypoplasia and/or coloboma) [31,60,62,69,69,70,84,87,92]. Abnormalities of visual function were described in 11% of the articles [62,69,80,82,84], as well as extrinsic ocular motility (nystagmus and strabismus) [60,62,70,80]. Abnormalities in the anterior segment (cataract and glaucoma) appeared in 9% of the articles [70,87,94] and refractive error (myopia, hyperopia and astigmatism) in 4% [62,87].

Lesions in other organ systems were also observed, to a lesser extent: cardiovascular system, [19], genitourinary system (ambiguous genitalia [75] and bilateral cryptorchidism [40,84]), and gastrointestinal system (dysphagia [19,71]) (Table 2). Other clinical features included small for gestational age (SGA) in six studies [19,23,69,84,85,90], hearing abnormalities in four [19,59,63,87], and unilateral diaphragmatic paralysis in three articles [40,70,73] (Table 2).

Three articles described the association between congenital ZIKV infection, microcephaly, and other adverse pregnancy outcomes [19,23,58]. One case-control study demonstrated the association between microcephaly and ZIKV in neonates tested by Zika virus-specific IgM and quantitative RT-PCR in serum and cerebrospinal fluid. In addition to microcephaly, this study detected congenital malformations through abnormal brain findings on imaging tests in children with virus in the serum or cerebrospinal fluid and serum only [23]. One cohort study described adverse pregnancy outcomes, including cases of fetal loss in every trimester [19]. Another cohort study identified an association between ZIKV infection and CNS (central nervous system) anomalies, cerebral hyperechogenicity, and intrauterine fetal death [58] (Table 3).

Imaging tests were used in 31 (67%) of the articles reviewed to detect altered signs potentially associated with Zika virus syndrome: ultrasound (80%), computed tomography (55%), magnetic resonance imaging (29%), and transfontanelar ultrasound (23%). Fourteen (45%) and 12 (39%) studies used three and two imaging methods, respectively (Table 4). The most frequent signs observed in imaging tests were located in the central nervous system: ventriculomegaly described in 35 articles [19,23,45,58,64–71,73,74,76,78–80,82–85,87,89–94,96,98,100, 101]; parenchymal or cerebellar calcification (n = 33) [19,23,45,59,62–71,73,74,76,78–80,82–85,87,91,93,94,96–98,100,101]; microcephaly (n = 32) [19,40,45,58,59,63,64,66,67,70,73,74,76,79,82–87,89,91,93,94,96–101]; hypoplasia or atrophy of the cerebral cortex, cerebellum, or brainstem (n = 29) [19,45,62–67,69–71,73,74,76,79,80,82,84,85,87,89,91,93–96,98,100,101]; abnormal cortical formation (n = 27) [19,23,40,58,65–67,69–71,73,74,76,78–82,84,87,90–93,100,101]; corpus callosum anomaly (n = 18) [45,58,62,65,67–70,74,76,79,80,82,91,92,94,96,98]; and hydrocephalus (n = 3) [40,63,86] (Table 4). Cardiomegaly and diaphragmatic paralysis were detected by chest x-ray [102].

Additional lesions observed with imaging tests were: polyhydramnios or oligohydramnios [19,58,64,70,74,90,100] and intrauterine growth restriction [19,63,64,82,85,91,97], described in seven articles. As the clinical examination indicated, osteoskeletal lesions were also frequent: arthrogryposis (n = 6) [19,40,64,70,74,86] and clubfoot [19,64,79] (n = 3). Other organ systems

**Table 2. Clinical signs of children exposed to ZIKV in pregnancy.**

| Clinical signs | | | Min[δ] | Max[£] | n (%) | Studies N = 46 |
|---|---|---|---|---|---|---|
| **Neurological abnormalities** | Microcephaly | | 1 | 87 | 21 (46) | [19,23,40,59,60,62,63,65,69,71,73,76,78,80,84,87,90,92,93,99,100] |
| | Hypertonicity | | 1 | 58 | 8 (17) | [19,62,63,71,80,87,93,94] |
| | Seizures | | 1 | 87 | 7 (15) | [19,40,62,70,71,73,84] |
| | Neurological crying/irritability during first months of life | | 1 | 58 | 6 (13) | [19,71,80,83,87,94] |
| | Hyperexcitability/hyperreflexia | | 1 | 58 | 5 (11) | [19,70,80,84,87] |
| | Asymmetrical tonic neck reflex | | 1 | 13 | 4 (9) | [70,71,83,87] |
| | Clenched fists | | 12 | 58 | 2 (4) | [19,80] |
| | Distal tremors | | 12 | 58 | 2 (4) | [19,80] |
| | Centrally decreased muscle tone (upper extremities) | | 1 | 58 | 2 (4) | [19,84] |
| | Neurological impairment | | 1 | 58 | 2 (4) | [19,83] |
| | Encephalocele | | 1 | 26 | 2 (4) | [59,99] |
| | Altered visual fixation and pursuit | | 2 | 13 | 2 (4) | 16–17 |
| | Abnormal posturing | | | | 1 (2) | [19] |
| | Altered motor reflexes | | | | 1 (2) | [19] |
| | Hemiparesis | | | | 1 (2) | [19] |
| | Hypoactivity | | | | 1 (2) | [19] |
| | Hydrocephalus | | | | 1 (2) | [86] |
| | Cortical blindness | | | | 1 (2) | [84] |
| **Osteoskeletal abnormalities** | Arthrogryposis | | 1 | 87 | 12 (26) | [40,61,69,71,73,75,80,85–87,90,92] |
| | Hip dysplasia | | 13 | 58 | 2 (4) | [19,71] |
| | Clubfoot | | 1 | 13 | 2 (4) | [85,87] |
| | Knee fovea | | | | 1 (2) | [19] |
| | Cortical thumb | | | | 1 (2) | [19] |
| | Elbow fovea | | | | 1 (2) | [19] |
| | Polydactyly | | | | 1 (2) | [19] |
| | Hand contractures / Camptodactyly | | | | 1 (2) | [80] |
| | Feet malposition / contractures | | | | 1 (2) | [80] |
| | Prominent calcaneus | | | | 1 (2) | [80] |
| **Ophthalmic abnormalities** | Posterior segment abnormalities | Retinal abnormalities & | 1 | 16 | 10 (22) | [60,62,69–71,73,78,84,87,92] |
| | | Optic nerve abnormalities[#] | 1 | 15 | | [60,62,69,70,78,84,87,92] |
| | Abnormal visual function | | 1 | 32 | 5 (11) | [62,63,71,80,84] |
| | Extrinsic eye motility | Strabismus | ¨ | 30 | 5 (11) | [62,70,80] ¨ |
| | | Nystagmus | 6 | 9 | | [60,62] |
| | Anterior segment abnormalities | Cataract | 1 | 1 | 4 (9) | [70,98] |
| | | Glaucoma | 1 | 1 | | [87,92] |
| | Refractive error | Astigmatism | | | 2 (4) | [62] |
| | | Myopia | | | | [62,87] |
| | | Hyperopia | | | | [62] |

(*Continued*)

**Table 2.** (Continued)

| Clinical signs | | Children exposed to ZIKV | | | Studies N = 46 |
|---|---|---|---|---|---|
| | | Min[δ] | Max[£] | n (%) | |
| Abnormalities in other systems | Morphological changes of the head Φ | 1 | 87 | 10 (22) | [19,40,65,69,71,73,80,83,84,87] |
| | Small for gestational age (SGA) | 1 | 58 | 6 (13) | [19,23,69,84,85,90] |
| | Overriding sutures or closed fontanels | 1 | 13 | 5 (11) | [70,71,80,83,84] |
| | Hearing abnormalities | 1 | 58 | 4 (9) | [19,59,63,87] |
| | Unilateral diaphragmatic paralysis | 3 | 87 | 3 (7) | [40,70,73] |
| | Dystonic movement | 32 | 58 | 2 (4) | [19,62] |
| | Dysphagia | 13 | 58 | 2 (4) | [19,71] |
| | Bilateral cryptorchidism. | 1 | 3 | 2 (4) | [40,84] |
| | Large for gestational age (LGA) | | | 1 (2) | [19] |
| | Head lag | | | 1 (2) | [19] |
| | Failure to gain weight | | | 1 (2) | [19] |
| | Sacral dimple | | | 1 (2) | [19] |
| | Congenital heart disease | | | 1 (2) | [19] |
| | Dyskinesia | | | 1 (2) | [71] |
| | Increased deep tendon reflexes | | | 1 (2) | [80] |
| | Multiple dimples | | | 1 (2) | [80] |
| | Ambiguous genitalia | | | 1 (2) | [75] |
| | Coronal hypospadias | | | 1 (2) | [84] |

[δ]: minimum number of children investigated per article.

[£]: maximum number of children investigated per article.

[&]: Pigmented, chorioretinal atrophy and/or coloboma

[#]: Pallor, atrophy, increased excavation, hypoplasia and/or coloboma

[¨]: No information on number of children

Φ verlapping cranial sutures, prominent occipital bone, excess nuchal skin, craniofacial disproportion with depression of frontal and parietal bones

were involved, although less frequently: gastrointestinal (intestinal hyperechogenicity (n = 2) [58,68], liver calcifications and hepatomegaly (n = 2) [19,58], cardiovascular (tachyarrhythmia and cardiomyopathy (n = 1) [58], respiratory (hydrothorax (n = 1), and genitourinary (genitourinary tract anomaly (n = 2) [40,58] (Table 4).

The following other lesions were observed on imaging tests in one study [19] and can help elucidate the natural history of Zika virus infection: abnormal middle cerebral artery, abnormal umbilical artery flow, and placental insufficiency (Table 4).

Eight studies [40,58,68,81,84,90,93,103] reported the placenta's characteristics, especially in cases of fetal death. There were calcifications in five articles [40,68,81,93,103], and vascular alterations such as villous infarction [40,81,93] and scattered intervillous thrombi [84,93].

Seven articles reported data on miscarriage and autopsy in neonatal and fetal death: microcephaly (n = 6) [40,45,75,81,90,97], calcifications (n = 4) [40,75,81,90], CNS hypoplasia (n = 4) [40,45,75,81], ventriculomegaly (n = 3) [40,75,81], arthrogryposis (n = 5) [40,75,81,90,97], and pulmonary hypoplasia (n = 2) [40,75,90] (Table 5).

Neurological signs and symptoms were the ones most frequently described in both the imaging tests (Table 4) and autopsies (Table 5).

**Table 3. Clinical signs and adverse pregnancy outcomes associated with Zika Virus (ZIKV) exposure in pregnancy.**

| Outcomes | OR/RR | Adjusted OR | Study | Country |
|---|---|---|---|---|
| Microcephaly /ZIKV detected in serum or cerebrospinal fluid samples | 55.5 (8.6-∞) | 59.2 (9-∞)* | [23] | Brazil |
| | | 55,6 (8.5-∞)** | | |
| Microcephaly /abnormal brain imaging findings | 113.3 (14.5-∞) | | | |
| Microcephaly /normal brain imaging findings | 24.7 (2.9-∞) | | | |
| Central Nervous System anomaly | 2.11 (1.18–4.13) | | [58] | French Guiana |
| Corpus callosum anomaly | 2.21 (1.08–5.26) | | | |
| Cerebral hyperechogenicity | 3.98 (1.48–11.49) | | | |
| Intrauterine fetal death | 3.98 (1.09–15.17) | | | |
| Adverse pregnancy outcomes | 4.03 (1.96–8.32) | | [19] | Brazil |

* maternal education

** maternal age

## Discussion

In our review, the main signs and symptoms associated with congenital Zika virus syndrome were microcephaly, parenchymal or cerebellar calcifications, ventriculomegaly, CNS hypoplasia or atrophy; arthrogryposis; ophthalmic findings mainly focal pigmentary retina mottling, chorioretinal atrophy and/or coloboma, pallor, atrophy, increased excavation, hypoplasia and/or coloboma of optic nerve and abnormal visual function and low birthweight for gestational age.

Microcephaly was the most frequent sign found in neonates exposed to ZIKV during pregnancy among the articles in our review [28,38–41]. Forty one studies reported microcephaly [19,23,40,45,58–60,62–67,69,70,70,71,73–76,79–87,89–94,96–101]. Other studies, using different methodologies, have reported similar results [104–106], including increased prevalence of microcephaly among children exposed to ZIKV in other systematic review [47] when compared to the prevalence in other studies of unexposed children in Brazil and Europe [107,108].

Parenchymal or cerebellar calcification were described in 37 studies of our review [19,23,40,45,59,62–71,73–76,78–85,87,90,91,93,94,96–98,100,101] and also in other review which reported a prevalence of 42.6% [47]. These findings may also be found in congenital infections known by the mnemonic TORCH (Toxoplasmosis, Rubella, Cytomegalovirus (CMV) and Herpes. However, the distribution of intracranial calcifications differs in each congenital infection: typically larger, denser and subcortical in congenital ZIKV, punctate and periventricular or cortical in CMV, diffuse and widely distributed in congenital toxoplasmosis and at basal ganglia in rubella [41,109–111]. Besides that it should be noted that some neurological changes (calcifications and cerebral atrophy) in congenital Zika syndrome are similar to other syndromes of infectious and also genetic etiology, such as Aicardi-Goutières syndrome [112].

Ventriculomegaly was described in 32 studies reviewed [19,23,45,56,56,58,63–74,76,78–80,82–84,87–94,96,98,100], an important sign regarding the risk of developing hydrocephalus and the need for surgical intervention for ventriculoperitoneal shunt, which was also observed

**Table 4. Signs detected on imaging tests of children exposed to Zika Virus (ZIKV) in pregnancy.**

| | Signs | Type of test* | Children exposed to ZIKV | | Studies N = 46 |
|---|---|---|---|---|---|
| | | | Min[ð] | Max[£] | n (%) | |
| Neurological alterations | Ventriculomegaly/ increased fluid spaces | US, TU, CT, MRI | 1 | 81 | 35 (76) | [19,23,45,58,64–71,73,74,76,78–80,82–85,87,89–94,96,98,100,101] |
| | Parenchymal or cerebellar calcification | US, TU, CT, MRI | 1 | 81 | 33 (72) | [19,23,45,59,62–71,73,74,76,78–80,82–85,87,91,93,94,96–98,100,101] |
| | Microcephaly | US, TU, CT, MRI | 1 | 79 | 32 (70) | [19,40,45,58,59,63,64,66,67,70,73,74,76,79,82–87,89,91,93,94,96–101] |
| | Hypoplasia or atrophy of cerebral cortex, cerebellum, brainstem | US, TU, CT, MRI | 1 | 41 | 29 (63) | [19,45,62–67,69–71,73,74,76,79,80,82,84,85,87,89,91,93–96,98,100,101] |
| | Abnormal cortical formation (encephalomalacic changes, abnormal gyration, lissencephaly) | US, TU, CT, MRI | 1 | 81 | 27 (59) | [19,23,40,58,65–67,69–71,73,74,76,78–82,84,87,90–93,100,101] |
| | Corpus callosum anomaly | US, TU, CT, MRI | 1 | 32 | 18 (39) | [45,58,62–70,74,76,79,80,82,91,92,94,96,98] |
| | Cysts or pseudocysts | US, TU, MRI | 1 | 41 | 8 (17) | [19,63,68,76,87,88,96,101] |
| | Increased cisterna magna | US, CT, MRI | 1 | 41 | 7 (15) | [19,65,69,85,93,96,98] |
| | Fetal Dandy-Walker malformation | US, TU, CT, MRI | 1 | 30 | 3 (7) | [64,73,90] |
| | Hemorrhage | US, TU, MRI | 1 | 41 | 3 (7) | [19,45,96] |
| | Hydrocephalus | US, MRI | 1 | 8 | 3 (7) | [40,63,86] |
| | Posterior fossa anomaly | US, TU | 14 | 27 | 2 (4) | [58,82] |
| | Cerebellar abnormalities | US, MRI | 17 | 24 | 2 (4) | [78,79] |
| | Anencephaly | US | 2 | 4 | 2 (4) | [74,95] |
| | Lenticulostriate vasculopathy | TU | 2 | 2 | 2 (4) | [87,88] |
| | Supratentorial dilatation | US, TU, MRI | | | 1 (2) | [19] |
| | Ischemic parenchymal lesions | US, MRI | | | 1 (2) | [19] |
| | Abnormal middle cerebral artery | US | | | 1 (2) | [19] |
| | Brachycephaly | US | | | 1 (2) | [19] |
| | Cerebral hyperechogenicity | US, TU | | | 1 (2) | [58] |
| | Abnormal pons | US | | | 1 (2) | [82] |
| | Polymalformative syndrome (encephalocele, anophthalmia, arthrogryposis, fetal hydrops) | US | | | 1 (2) | [82] |
| | Holoprosencephaly | US | | | 1 (2) | [74] |
| | Schizencephaly | US | | | 1 (2) | [95] |
| | Hydranencephaly | US | | | 1 (2) | [97] |
| | Encephalocele | US | | | 1 (2) | [99] |
| | Colpocephaly | TU, MRI | | | 1 (2) | [91] |
| Alterations in other systems | Polyhydramnios/oligohydramnios | US | 1 | 41 | 7 (15) | [19,58,64,70,74,90,100] |
| | Intrauterine growth restriction | US, TU, CT, MRI | 1 | 41 | 7 (15) | [19,63,64,82,85,91,97] |
| | Arthrogryposis | US, TU, CT, MRI | 1 | 41 | 6 (13) | [19,40,64,70,74,86] |
| | Clubfoot | US, TU, CT, MRI | 17 | 41 | 3 (7) | [19,64,79] |
| | Premature closure of fontanelle | CT, MRI | 1 | 6 | 3 (7) | [93,65,91] |
| | Hepatomegaly/liver calcifications | US, TU, CT, MRI | 8 | 27 | 3 (7) | [58,63] |
| | Liver/spleen echogenicity | US | 27 | 41 | 2 (4) | [19,58] |
| | Intestinal hyperechogenicity | US | 4 | 27 | 2 (4) | [58,68] |
| | Small for gestational age (SGA) | US, TU | 3 | 27 | 2 (4) | [40,58] |
| | Genitourinary tract anomaly | US | 3 | 27 | 2 (4) | [40,58] |
| | Subcutaneous edema | US | 1 | 1 | 2 (4) | [96,97] |
| | Hydrothorax | US | 1 | 1 | 2 (4) | [96,97] |
| | Fetal macrosomia | US | | | 1 (2) | [19] |
| | Abnormal umbilical artery flow | US | | | 1 (2) | [19] |
| | Placental insufficiency | US | | | 1 (2) | [19] |
| | Tachyarrhythmia | US | | | 1 (2) | [58] |
| | Cardiomyopathy | US | | | 1 (2) | [58] |
| | Hyperechogenicity of aortic valve, mitral valve, and aortic root | US, TU, CT, MRI | | | 1 (2) | [64] |
| | Thymic calcifications | US, TU, CT, MRI | | | 1 (2) | [64] |
| | Ascites | US | | | 1 (2) | [97] |

*US: pelvic or obstetric ultrasound; TU: transfontanelar ultrasound: CT: computed tomography, MRI: magnetic resonance imaging.

[ð]: minimum number of children investigated per article.

[£]: maximum number of children investigated per article.

in other studies [47,113]. In our study, hydrocephalus was associated with severe signs and symptoms, such as: epilepsy, motor dysfunction, cognitive dysfunction, arthrogryposis (one-third of cases), and visual and hearing disorders [40,66,82,86,95,101], in agreement with Linden (2019) [113], who identified severe neurological disorders and cognitive dysfunction in one-third of children who progressed with hydrocephalus.

**Table 5. Autopsy findings of children exposed to Zika Virus (ZIKV) in pregnancy.**

| Autopsy findings | | Children exposed to ZIKV | | | Studies N = 46 |
|---|---|---|---|---|---|
| | | Min[ð] | Max[£] | n (%) | |
| Neurological findings | Microcephaly | 1 | 10 | 6 (13) | [40,45,75,81,90,97] |
| | Calcifications | 1 | 10 | 4 (9) | [40,75,81,90] |
| | CNS hypoplasia | 1 | 10 | 4 (9) | [40,45,75,81] |
| | Ventriculomegaly | 3 | 10 | 3 (7) | [40,75,81] |
| | Inflammatory infiltrate in leptomeninges and brain | 1 | 7 | 3 (7) | [40,45,75] |
| | Vascular congestion in leptomeninges or brain | 3 | 7 | 2 (4) | [40,75] |
| | Anterior spinal horn cell loss | | | 1 (2) | [81] |
| | Pachygyria | | | 1 (2) | [75] |
| | Cerebral gliosis | | | 1 (2) | [75] |
| | Lissencephaly | | | 1 (2) | [40] |
| | Holoprosencephaly | | | 1 (2) | [40] |
| | Cerebral necrosis | | | 1 (2) | [40] |
| | Hydrocephalus | | | 1 (2) | [40] |
| | Malformation of cortical development with agyria | | | 1 (2) | [90] |
| | Dandy-Walker syndrome | | | 1 (2) | [90] |
| Findings in other systems | Arthrogryposis | 1 | 10 | 5 (11) | [40,75,81,90,97] |
| | Pulmonary hypoplasia | 1 | 3 | 3 (7) | [40,75,90] |
| | Pulmonary hemorrhage | 3 | 7 | 2 (4) | [40,75] |
| | Virus detected in fetal tissues | 3 | 7 | 2 (4) | [40,63] |
| | Hydropic degeneration of liver | 1 | 7 | 2 (4) | [75,90] |
| | Neurogenic muscle atrophy | | | 1 (2) | [40] |
| | Inflammatory infiltrate in lung | | | 1 (2) | [75] |
| | Liver apoptosis | | | 1 (2) | [75] |
| | Liver steatosis | | | 1 (2) | [75] |
| | Intrauterine growth restriction | | | 1 (2) | [75] |
| | Ventricular septal defect | | | 1 (2) | [90] |

[ð]: minimum number of children investigated per article.

[£]: maximum number of children investigated per article.

Congenital infections such as those involving *Toxoplasma gondii* and cytomegalovirus have also been associated with serious brain alterations such as calcifications and ventriculomegaly [114,115]. However, unlike these other infections, ZIKV infection is known to be associated with severe microcephaly, with partially collapsed skull, thin cerebral cortex with subcortical calcifications, macular scars, and retinal changes as a result of the important viral tropism for fetal neural and ocular progenitor cells [116]. In addition, congenital contractures (bone deformities) are also associated with ZIKV infection during pregnancy, although observed less frequently [117]. In short, these findings can also occur in similar congenital syndromes associated with other infectious diseases, such as toxoplasmosis, syphilis, varicella, parvovirus

B1, rubella, cytomegalovirus, and herpes simplex [23,42,58,118] but the signs and symptoms presented by CZS seem to be more serious.

The osteoskeletal system was the second most frequently altered organ system in newborns exposed *in uterus* to ZIKV and arthrogryposis was described in 18 studies reviewed [19,40,61,64,69–71,73–75,80,81,85–87,90,92,97], as observed by other studies, mainly in more severe cases [15,47,79,113].

Anatomical [60,62,69–71,73,78,84,87,92] and functional [62,63,71,80,84] changes of the eye were described in 12 articles, such as posterior and anterior segment abnormalities, extrinsic eye motility and abnormal visual function. Many of these anatomical changes may impact the child's visual function in the future, since they affect prime areas of the eye. Importantly, since the child's first contact with the environment is through eyesight, children with visual impairment commonly experience delayed neuropsychomotor development, which, alongside the syndrome's other signs and symptoms, hinder the child's ability to integrate with his or her surroundings [119]. These ocular findings were described also in other studies descripted [47,120–122].

TORCH infections have also been associated with serious ophthalmic alterations. In congenital toxoplasmosis infection, chorioretinal lesions are usually bilateral and can present with active lesions or regressed scar [60,123]. Congenital cytomegalovirus and herpes virus infections can also manifest with active ocular inflammation. In Zika virus infection, however, there has been no active ocular inflammation cases reported so far [123]. There is usually not pigmentary mottling seen outside of areas of chorioretinal atrophy in congenital toxoplasmosis infection, as seen in zika virus infection [60]. In congenital rubella infection, the pigment mottling is usually diffused compared with the focal pigment mottling seen in zika virus infection [60]. Congenital cytomegalovirus can present with chorioretinal lesions similar to congenital toxoplasmosis but less heavily pigmentated and with pale or small optic disc [124,125]. Optic nerve hypoplasia, commonly seen in ZIKV congenital infection, is rarely seen in rubella, toxoplasmosis and herpes congenital infections [60].

The pathophysiology of ZIKV infection and the mechanism of the virus' passage across the placental barrier are still under investigation [48]. The exact timing of placental and fetal infection in relation to maternal viremia is also still unclear, as is the correlation between prolonged viremia and the development of congenital Zika syndrome [48]. Placental alterations were reported in eight studies [40,58,68,81,84,90,93,103], but the role of placental viral infection in determining the syndrome's severity has not been determined to date, as stated by other authors who found mild and nonspecific pathological findings in the placenta in pregnancies with ZIKV infection [126–129].

The effects of ZIKV on the fetus are more frequent and severe when maternal infection occurs in the first and second trimesters of pregnancy, resulting in: spontaneous abortion; therapeutic abortion due to major congenital malformation; various congenital malformations; fetal death; and neonatal and post-neonatal deaths [19,42,59,60], as corroborated by other studies [107,130–133]. Notably, the total number of abortions in Brazil, the epicenter of ZIKV cases, may be underestimated, since therapeutic abortion is prohibited by law, even though a recent study did not find an increase in hospitalizations due to complications of abortion during the epidemic [77]. In addition to adverse pregnancy outcomes and deaths, our review presents evidence that early maternal infection during pregnancy is associated with higher likelihood of congenital abnormalities such as microcephaly, ophthalmic lesions, osteoskeletal malformations, and various brain alterations [58,88] as described in other studies [19,129,130,133–141].

Among the imaging tests performed, obstetric ultrasound was the most frequent, in addition to transfontanelar ultrasound (TU), computed tomography (CT), and magnetic resonance

imaging (MRI) of the newborn (Table 4). Transfontanelar ultrasound, CT, and MRI were reported less frequently, possibly due to the limited access to those exams in some settings. Obstetric ultrasound proved to be useful for tracking ZIKV-related brain injury in prenatal care in pregnant women exposed to ZIKV. Although less accurate than CT and MRI it is available in many health facilities [142,143].

One point to highlight in our review is the anatomopathological analysis. It was possible to notice either the severity of the systemic impairment of the cases that died, especially those whose maternal infection occurred in the first trimester of pregnancy and the correspondence of clinical and anatomopathological findings.

Low birthweight and intrauterine growth restriction were common findings in the studies in the current review [19,63,64,69,82,85,89–91], representing a risk for children exposed to ZIKV during pregnancy. A recent study found that prevalence of Low birthweight in infants with CZS was more than four times that of the overall sample of live births [144].

Gastrointestinal disorders were reported [19,58,64,75,90], including liver disorders [58,68], intestinal alterations, and dysphagia [68,71]. Dysphagia can increase the risk of bronchoaspiration, resulting in aspiration pneumonia or death from asphyxiation, requiring gastrostomy in the affected infants [113,145]. Structural changes in the genitourinary system were reported [40,58,75,84] and were also described by Villamil-Gómez (2019) in fetal autopsies [146]. Bladder impairment and possible kidney damage such as very low bladder capacity, bladder hyperactivity with increased consistency, high bladder pressure during the filling phase, and high postvoid residual volume (PVR) and/or recurrent urinary tract infection were reported in older children [147]. Congenital cardiovascular anomalies were identified in children exposed to ZIKV such as cardiomyopathy, hyperechogenicity of aortic valve, mitral valve, and aortic root [19,58,64,100]. Other studies have also described cardiological symptoms and imaging findings such as complex congenital heart disease, echocardiographic abnormalities, and cardiac overload [148–151].

Pulmonary alterations were less common, mostly detected by autopsy and thus tending to be more severe [40,75,90]. Other abnormalities such as hydrothorax [96,97] and diaphragmatic paralysis [40,73,98] were reported. A recent study investigating the cause of diaphragmatic paralysis in three infants with CZS reported phrenic nerve dysfunction detected by electromyography [113]. Hearing abnormalities were reported in some studies [19,59,63,87], highlighting the need for specific tests for early detection, since they can further impair the child's interaction with the environment and affect neurodevelopment, as described by Leal *et al.* (2016) [152].

Congenital Zika syndrome and all its effects on children according to age group still need to be evaluated in light of new findings. Our review supported the characterization of the syndrome in infants up to six months of age since the initial reporting of cases. Recent studies have identified neurological disorders such as delay in neurodevelopment, mainly in the domain of language in children exposed to ZIKV who were asymptomatic at birth [133,153]. Such findings are relevant, since there is evidence that ZIKV can continue to replicate in the infant's brain after birth [154], and that cerebral growth of infants exposed during pregnancy can decelerate, even after birth [69].

Cohort studies are needed to characterize the syndrome's signs and symptoms according to age group in order to assess the impact on children's cognitive development. Recent studies indicate that neurological changes can negatively impact the children's motor and cognitive levels, with serious consequences for the social life of the patients and their families. Most of these children display severe motor dysfunction [155,156], delayed functional performance [132], difficulties in eating and sleeping, visual and auditory abnormalities, seizures [156], swallowing dysfunction, movement abnormalities, and epilepsy [153]. Complications

associated with respiratory infections, dysphagia, and epilepsy can be fatal for the most severely affected infants [97,64,135].

Congenital Zika syndrome is a serious public health problem, both because of the clinical severity of the cases and the extent of functional impairment. Absence of signs and symptoms at birth in exposed infants does not rule out their appearance later in childhood, thus highlighting the importance of structuring healthcare networks for comprehensive monitoring and care of these children. The high prevalence of asymptomatic cases at birth (65 to 83%) [5,157] can delay the identification of the association between congenital syndromes (with or without late onset) and ZIKV infection in the mother during pregnancy. Efforts are needed to provide care and support for all the needs of children with congenital Zika syndrome and their families, as well as effective organization of healthcare and social services.

The ZIKV reemergence in regions with autochthonous transmission and the introduction of the virus in areas with established vector mosquito infestation may increase the risk of the development of congenital Zika syndrome in all regions of the world, especially in Africa, the Americas, Southeast Asia and the Western Pacific [158].

In the absence of effective vaccines, licensed to date [161,162], and considering the complexity of effective vector control, innovative intersectoral strategies that transcend exclusive vector chemical control actions should be incorporated in areas with viral circulation [159–164]. Prevention strategies such as the use of repellents by pregnant women and condoms by sexual partners, both for travelers to regions with ZIKV circulation, and for residents of risk areas, as well as the implementation of strict screening protocols in the donation system of blood should be implemented. In addition, pregnant women and those intending to become pregnant may be advised to avoid unnecessary travel to endemic regions [165].

One limitation to the study was the low number of comparative clinical studies. The inclusion of case series and case reports, with lower levels of scientific evidence, is justified for the investigation of new and rare diseases such as congenital Zika syndrome, since evidence based on clinical case reports is necessary to understand the natural history of a new illness [166]. We chose to include only studies with laboratory confirmation of cases, seeking accurate estimates of maternal-fetal transmission and risk of symptomatic congenital infection. This criterion excluded other studies published during the period analyzed.

## Conclusion

The review of clinical, imaging, and anatomopathological findings allowed the characterization of congenital Zika syndrome, especially in children born to mothers with laboratory-confirmed ZIKV infection during pregnancy.

Our results show that the congenital Zika syndrome encompasses several malformations mainly in the neurological, osteoskeletal, and visual systems, although the syndrome can affect other organ systems. ZIKV infection is also associated with several adverse pregnancy outcomes, including fetal loss.

We highlight the following findings: (a) in the neurological system: ventriculomegaly, parenchymal or cerebellar calcification, microcephaly, hypoplasia or atrophy of the cerebral cortex, cerebellum, and brainstem, abnormal cortical formation, corpus callosum anomaly, hydrocephalus, hypertonicity, and seizures; (b) in the osteoskeletal system: arthrogryposis and clubfoot; and (c) in the visual system: ophthalmic changes in the posterior and anterior segments and abnormal visual function.

Congenital Zika syndrome also impacts the health system and the families' daily lives, requiring collaboration between levels of healthcare and inter-sector cooperation aimed at comprehensive care for these children. The current review's findings are thus essential for

understanding the disease profile in patients and to assist the development and updating of more specific clinical protocols.

## Supporting information

**S1 Table. Systematic review's PICO.**
(PDF)

**S1 Fig. Checklist PRISMA.**
(PDF)

**S1 Appendix. International prospective register of systematic reviews–PROSPERO.**
(PDF)

**S2 Appendix. Quality score assessment.**
(PDF)

**S3 Appendix. Search strategy.**
(PDF)

**S1 File. Neurological disorders; Osteoskeletal abnormalities; Ophthalmic abnormalities; Abnormalities in other systems.**
(PDF)

## Acknowledgments

We are grateful for support from the Coordination for the Improvement of Higher Education Personnel (CAPES).

## Author Contributions

**Conceptualization:** Danielle A. Freitas, Reinaldo Souza-Santos.

**Data curation:** Danielle A. Freitas, Wagner B. Barros.

**Formal analysis:** Danielle A. Freitas, Reinaldo Souza-Santos, Liege M. A. Carvalho, Luiza M. Neves, Mayumi D. Wakimoto.

**Funding acquisition:** Danielle A. Freitas.

**Investigation:** Danielle A. Freitas, Reinaldo Souza-Santos, Wagner B. Barros, Luiza M. Neves, Mayumi D. Wakimoto.

**Methodology:** Danielle A. Freitas, Reinaldo Souza-Santos, Wagner B. Barros, Mayumi D. Wakimoto.

**Project administration:** Danielle A. Freitas, Reinaldo Souza-Santos, Mayumi D. Wakimoto.

**Resources:** Danielle A. Freitas, Reinaldo Souza-Santos, Mayumi D. Wakimoto.

**Software:** Danielle A. Freitas, Reinaldo Souza-Santos, Mayumi D. Wakimoto.

**Supervision:** Danielle A. Freitas, Reinaldo Souza-Santos, Patrícia Brasil, Mayumi D. Wakimoto.

**Validation:** Danielle A. Freitas, Reinaldo Souza-Santos, Mayumi D. Wakimoto.

**Visualization:** Danielle A. Freitas, Reinaldo Souza-Santos, Mayumi D. Wakimoto.

**Writing – original draft:** Danielle A. Freitas, Reinaldo Souza-Santos, Mayumi D. Wakimoto.

**Writing – review & editing:** Danielle A. Freitas, Reinaldo Souza-Santos, Liege M. A. Carvalho, Patrícia Brasil, Mayumi D. Wakimoto.

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
