## [Decision Letter · Decision Letter 0]

13 Aug 2020

PONE-D-20-18691

Congenital Zika Syndrome: a systematic review

PLOS ONE

Dear Dr. Freitas,

Thank you for submitting your manuscript to PLOS ONE. After careful consideration, we feel that it has merit but does not fully meet PLOS ONE’s publication criteria as it currently stands. Therefore, we invite you to submit a revised version of the manuscript that addresses the points raised during the review process.

ACADEMIC EDITOR: Editorial comments are appended below.

We look forward to receiving your revised manuscript.

Kind regards,

Aya Mousa, PhD

Academic Editor

PLOS ONE

Journal Requirements:

Additional Editor Comments (if provided):

Please address the comments by the reviewers carefully and provide a thoroughly revised version of the manuscript, with particular attention to the comments made by Reviewer 1 regarding the work being derivative and having no new information/ conclusions, as well as the grammatical/ typographical errors in the manuscript.

Inclusion of any citations requested by reviewers is not mandatory and is up to the authors' discretion.

Further Editorial comments below:

- The introduction is somewhat lengthy, consider making it more concise particularly pages 1-2.

- Where can this 'preestablished protocol' be found? Was it published or registered? If not, then unfortunately, you cannot claim it was preestablished.

- Line 144- a comprehensive search strategy does not necessary reduce publication bias since the databases used were traditional journal databases. Was grey literature explored?

- Page 7- Please include a PICO table with exclusion and inclusion criteria for the search as Table 1 in the methods.

- Page 7- The excluded studies are a bit unclear- why exclude research on vaccines and prevention if this might be epidemiological or clinical research with data pertaining to the question at hand? Same with molecular biology research- these criteria are vague and broad, it would be best to specify the study designs to be excluded and not the overall research area.

- Line 167- Data extraction is usually performed by 2 independent reviewers and cross-checked. This is a substantial limitation.

- Line 169- 'elaborated' is not the right word here.

- Line 174- 179- The way by which quality assessment was performed is confusing. Was this done at the study level or outcome level? It seems that for some studies this was at the study level assessing risk of bias and for others, this was done at the outcome level using the GRADE approach. And which items were selectively excluded from the GRADE measure? I think it would be best to find a more suitable measure to assess risk of bias in your studies rather than adapting an existing one to suit your types of studies.

- Line 204- A 'description was performed' is grammatically incorrect.

- Line 207- If quantitative data were available, why was a meta-analysis not performed?

- Line 215: The quality of articles was assessed with the MINORS instrument for all observational studies and with the Grading of Recommendations, Assessment, Development, and Evaluation (GRADE) approach- were both methods used for all studies? The way this is written is unclear.

- Line 218- If MINORS is used for non-comparative studies, how did you assess study-level quality for the 3 comparative studies? GRADE cannot be used for this purpose as it is an outcome-level assessment.

- Line 218- what is meant by 'most' did not report adequately.. how many?

- Line 220- 'care' reports?

- Line 221- What is a sufficiently long follow up defined as?

- Figure 2 is titled 'Quality assessment of comparative studies (MINORS)' whereas in line 176 you state that the first 8/12 items on the scale are for noncomparative studies. It appears the tool used here is inappropriate. Please use a more suitable tool.

- As directed by Reviewer 1, some of the table language is not in English- please correct and proof this (PAÍS? Mather?)

- The symbols used in Table 1 for diagnostic method are missing in some parts and difficult to follow- suggest including the words/ abbreviations instead or finding a better way of presenting this information.

- Line 263 onwards- please include referencing for the studies discussed (this is done in other sections e.g. line 325-330, but not here and should be consistent).

- It seems there are sufficient data for a meta-analysis.

- Line 334- Correct bracket/ parantheses

- Line 334-336 and onwards- Include references for the studies here and throughout.

- Line 396-400: Can this be presented in a table to show each sign and symptom and how many studies reported on this being associated with ZKV (and their references)?

- Line 401: how many studies reported on microcephaly?

- The terms ZIKV, ZKV and Zika virus or Zika syndrome are all used. Please use the abbreviation specified in the introduction and keep this consistent throughout

- Line 401-448: This entire section of the discussion just reiterates the results (as does most of the discussion). I would summarise and place into context- why is this important? What is new here?

- In agreement with Reviewer 1, more thought it needed to describe what this review adds to the literature and what the limitations of the literature are.

- What are the limitations of the review itself? You need a section describing the strengths and limitations, potential for bias in your results and issues with methodological rigour.

- The manuscript would be improved with more careful thought in the discussion and presenting results in light of study limitations as well as discussion of the implications of the findings. At present, the discussion just describes the results again and there is a need for further synthesis/ summarising and description of what this study adds to what is already known.

- The conclusion reads well.

Reviewers' comments:

Reviewer's Responses to Questions

**Comments to the Author**

1. Is the manuscript technically sound, and do the data support the conclusions?

Reviewer #1: Partly

Reviewer #2: Yes

2. Has the statistical analysis been performed appropriately and rigorously? 

Reviewer #1: N/A

Reviewer #2: N/A

3. Have the authors made all data underlying the findings in their manuscript fully available?

Reviewer #1: No

Reviewer #2: Yes

4. Is the manuscript presented in an intelligible fashion and written in standard English?

Reviewer #1: Yes

Reviewer #2: Yes

5. Review Comments to the Author

Reviewer #1: The authors conducted a systematic review of observational studies regarding congenital ZKV syndrome. While methods are valid and the authors cared to assess quality of studies and include only those with laboratorial confirmation of infection, the work is derivative. I could not identify what new information, conclusion or perspective this particular study brings to the literature.

Additionaly, some studies included are case reports with few patients, several have sample size n=1, which makes it hard to draw solid conclusions based on these studies. Therefore I am afraid that it does not merit enough to be published in the current state.

PLOS ONE Publication Criteria Assessment:

"1. The study presents the results of original research" - Yes.

"2. Results reported have not been published elsewhere." - Derivative study; the conclusions draw are similar to many other studies.

"3. Experiments, statistics, and other analyses are performed to a high technical standard and are described in sufficient detail." - Due to sample size concerns, statistics were not performed, several studies have sample sizes n=1. Results display is subpar, since tables are full of formatting errros and inadequacies (see minor suggestions below)

"4. Conclusions are presented in an appropriate fashion and are supported by the data." - Yes, but again check point 2.

"5. The article is presented in an intelligible fashion and is written in standard English." - Corrections should be made (see minor suggestions below)

"6. The research meets all applicable standards for the ethics of experimentation and research integrity." - Not applicable.

"7. The article adheres to appropriate reporting guidelines and community standards for data availability." - Not applicable.

# Minor suggestions

Page 19 line 374 - "There were calcifications in five articles" change to "Calcifications were reported in five studies"

Page 22 lines 450-451 - Unformatted reference? (ROBINSON et al., 451 2018)

Table 1 - In the caption, it should be "...characteristics of the study SAMPLE" not POPULATION

Table 1 - Words in Portuguese are scattered throughout the table (examples, País, imunohistoquimica)

Table 1 - change title of "POPULATION" column to "SAMPLE SIZE"

Table 1 - Correct typo in the column "MOTHER'S", not "MATHER'S"

Table 1 - What is "null" in Carvalho et al (2016) row, column CEPHALIC PERIMETER?

Table 1 - In DIAGNOSTIC METHOD(ZIKV) column some legend symbols are rendering as white squares. I suggest changing to letters or numbers if possible.

Table 1 - In English, decimal separators are points (.) not commas (,). Change all instances throughout Table 1

Figure 3 - Improve quality of this figure. The numeric axis is unreadable and the text in the quality assessment categories are truncated. Font color should be black, not grey.

Table 3 - Instead of "Associated factors" it should be "outcomes". Again, change "population" to "SAMPLE SIZE"

Table 3 - Why "associated factors" are repeated in this table? Reformat to reduce visual polution

Table 3 - Why some studies have the wording case/control and others have exposed/unexposed? Standardize nomenclature.

Table 3 - Why studies in the bottom part of the table 3 have contingency tables of case/control/whatever groups and the upper part does not?

Table 3 - Why "Not calculated"? If you have the contingency tables (as it seems to be the case) you could calculate yourselves.

References - Words in Portuguese (excluding the titles) are scattered throughout citations

Reviewer #2: Well written systematic review on Congenital Zika Syndrome (CZS).

The introduction could be strengthened to expand on international travelers.

Zika as arboviral disease in pregnancy should be briefly compared to other arboviral diseases in pregnancy, and this paper could be referred to:

Dengue, Zika and chikungunya during pregnancy: pre- and post-travel advice and clinical management.

Vouga M, Chiu YC, Pomar L, de Meyer SV, Masmejan S, Genton B, Musso D, Baud D, Stojanov M. J Travel Med. 2019 Dec 23;26(8):taz077. doi: 10.1093/jtm/taz077. PMID: 31616923 Free PMC article. Review.

The authors would do well highlighting how Zika could still re-emerge post 2016 outbreak in Asia and Africa where seroprevalence, and hence herd immunity, is not so high as in the Americas:

Zika in Angola and India.

Hamer DH, Chen LH. J Travel Med. 2019 Jun 11;26(5):taz012. doi: 10.1093/jtm/taz012.

Incidence of laboratory-confirmed Zika in Israeli travelers to Thailand: 2016-2019.

Leshem E, Lustig Y, Brosh-Nissimov T, Paran Y, Schwartz E. J Travel Med. 2019 Oct 14;26(7):taz057. doi: 10.1093/jtm/taz057.

The introduction should also briefly highlight that Guillain-Barre Syndrome is the other feared complication of Zika.

The methods section should highlight the need for a good case definition:

Zika virus infection in pregnancy: Establishing a case definition for clinical research on pregnant women with rash in an active transmission setting.

Ximenes RAA, Miranda-Filho DB, Brickley EB, Montarroyos UR, Martelli CMT, Araújo TVB, Rodrigues LC, de Albuquerque MFPM, de Souza WV, Castanha PMDS, França RFO, Dhália R, Marques ETA; Microcephaly Epidemic Research Group (MERG). PLoS Negl Trop Dis. 2019 Oct 7;13(10):e0007763. doi: 10.1371/journal.pntd.0007763. eCollection 2019 Oct. PMID: 31589611 Free PMC article.

Not sure whether I overlooked it, but the issue of asymptomatic Zika infections in pregnancy and their fetal outcomes needs to be highlighted.

The discussion should end with the need for bringing a vaccine to the market and underpin the hurdles and challenges to indeed achieve this.

6. PLOS authors have the option to publish the peer review history of their article (what does this mean?). If published, this will include your full peer review and any attached files.

Reviewer #1: No

Reviewer #2: No

---

## [Author Response · Author response to Decision Letter 0]

27 Oct 2020

Thanks for your comments.

Reviewer 1

The introduction is somewhat lengthy, consider making it more concise particularly pages 1-2.

The introduction section has been reduced as recommended

Where can this 'preestablished protocol' be found? Was it published or registered? If not, then unfortunately, you cannot claim it was preestablished.

The protocol of the systematic review was registered at PROSPERO under number CRD42020151754 on 10/27/2019 and is accessible at the link: https://www.crd.york.ac.uk/prospero/display_record.php?RecordID=151754

Line 144- a comprehensive search strategy does not necessary reduce publication bias since the databases used were traditional journal databases. Was grey literature explored?

In addition to conducting an extensive search of the available scientific literature, we also explored the grey literature, but no documents were retrieved according to the inclusion and exclusion criteria established as described in line 106. 

“Additionally, manual search was performed for bibliographic references of the selected articles and grey literature databases were also included to minimize publication bias [51–55]

Page 7- Please include a PICO table with exclusion and inclusion criteria for the search as Table 1 in the methods.

The elements and the search strategy used according to the acronym PICO are presented in the supplementary material. The description was included in the methods section, line 99.

We used PICO (population, intervention, comparison, outcome) as a search strategy tool as described in S1 Table. Line 108

Page 7- The excluded studies are a bit unclear- why exclude research on vaccines and prevention if this might be epidemiological or clinical research with data pertaining to the question at hand? Same with molecular biology research- these criteria are vague and broad, it would be best to specify the study designs to be excluded and not the overall research area.

The following types of publication were excluded: “editorials, letters, guidelines, and reviews.” (line 118). Studies were not included if they did not meet the inclusion criteria: “report data describing signs and symptoms of fetal growth impairment and altered development in children exposed to Zika virus during pregnancy, among laboratory-confirmed mothers and/or children <=2 years old”. (line 126) 

Line 167- Data extraction is usually performed by 2 independent reviewers and cross-checked. This is a substantial limitation.

Data extraction was performed by two independent reviewers (DFA and WB), and reviewed by the other (MDW) as described in line 125

Line 169- 'elaborated' is not the right word here

It was replaced by “made”. (line 137)

Line 174- 179- The way by which quality assessment was performed is confusing. Was this done at the study level or outcome level? It seems that for some studies this was at the study level assessing risk of bias and for others, this was done at the outcome level using the GRADE approach. And which items were selectively excluded from the GRADE measure? I think it would be best to find a more suitable measure to assess risk of bias in your studies rather than adapting an existing one to suit your types of studies.

Line 215: The quality of articles was assessed with the MINORS instrument for all observational studies and with the Grading of Recommendations, Assessment, Development, and Evaluation (GRADE) approach- were both methods used for all studies? The way this is written is unclear.

Line 218- If MINORS is used for non-comparative studies, how did you assess study-level quality for the 3 comparative studies? GRADE cannot be used for this purpose as it is an outcome-level assessment.

Line 218- what is meant by 'most' did not report adequately.. how many?

To assess the quality of case reports we used JBI (Joanna Briggs Institute) critical appraisal checklist for case reports, which we considered more suitable to our studies. (Line 145). MINORS was used for the following study designs: cohort, case-control, sectional and case series. Please see in Methods and in Results. 

Methods: …“Assessment of the methodological quality of the observational studies was based on the Methodological Index for Non-Randomized Studies (MINORS) [53]. The instrument consists of 12 items, the first eight being specific for noncomparative studies. To assess the quality of case reports, we used JBI (Joanna Briggs Institute) critical appraisal checklist for case reports [54]. (line 142)

Results … “Among the articles, more than 90% presented end points appropriate to their aim and consecutive inclusion of patients according to Minors. Less than 20% of the studies reported adequately on the following criteria: “loss to follow-up less than 5%”, “prospective calculation of the study size”, “an adequate control group”, “Contemporary groups”, “baseline equivalence of groups” and “adequate statistical analyses” (Fig 2). From all case reports 36 % did not describe patient´s history adequately. The other items were adequate in more than 90% of studies and two were not applicable (Fig 3).”

We also replaced figures 2 and 3. (line 163)

Line 204- A 'description was performed' is grammatically incorrect.

The sentence has been modified to: 

…“The studies were described according to country, study and sample characteristics, clinical examination, imaging tests, autopsy findings….” (Line 151)

Line 207- If quantitative data were available, why was a meta-analysis not performed?

It was not possible to perform a meta-analysis because most of the studies included were case reports and case series. The remaining four comparative studies presented different variables and outcomes.

Line 220- 'care' reports?

The word was misspelled, it was “case”. However, the text was excluded because we have modified the quality assessment method. 

Line 221- What is a sufficiently long follow up defined as?

The text was excluded because we have modified the quality assessment method. 

Figure 2 is titled 'Quality assessment of comparative studies (MINORS)' whereas in line 176 you state that the first 8/12 items on the scale are for noncomparative studies. It appears the tool used here is inappropriate. Please use a more suitable tool.

“Quality assessment of the studies included in the systematic review based on Methodological Index for Non-Randomized Studies (MINORS).”

MINORS is used to assess the quality of non-randomized studies. We used this method to evaluate cohort, case-control, cross-sectional and case series studies. The instrument has 12 items, the last four apply only to comparative studies.

As directed by Reviewer 1, some of the table language is not in English- please correct and proof this (PAÍS? Mather?)

We reviewed all the text of the manuscript. Portuguese terms were excluded.

The symbols used in Table 1 for diagnostic method are missing in some parts and difficult to follow- suggest including the words/ abbreviations instead or finding a better way of presenting this information.

The symbols were reviewed and some were replaced (line 185)

Line 263 onwards- please include referencing for the studies discussed (this is done in other sections e.g. line 325-330, but not here and should be consistent).

It seems there are sufficient data for a meta-analysis.

Line 334-336 and onwards- Include references for the studies here and throughout.

The references had been inserted in the tables only, but we included all the references cited in the text. (lines 215-217, 219-222, 227-231, 242-243, 245-246, 267-276, 278-285, 289-292, 300-302)

Line 334- Correct bracket/ parantheses

The brackets were removed.

“Lesions in other organ systems were also observed, to a lesser extent: cardiovascular system, [16], genitourinary system (ambiguous genitalia [72] and bilateral cryptorchidism [37,81]), and gastrointestinal system (dysphagia [16,68]) (Table 2). Other clinical features included small for gestational age (SGA) in six studies [16,20,66,81,82,87], hearing abnormalities in four [16,56,60,84], and unilateral diaphragmatic paralysis in three articles [37,67,70] (Table 2).” (line 241)

Line 396-400: Can this be presented in a table to show each sign and symptom and how many studies reported on this being associated with ZKV (and their references)?

We included tables presenting the most frequent signs associated with ZIKV reported in the studies, regardless the diagnostic method, in the supplementary material. 

The manuscript text in lines 396-400 was modified to “…In our review, the main signs and symptoms associated with congenital Zika virus syndrome were microcephaly, parenchymal or cerebellar calcifications, ventriculomegaly, CNS hypoplasia or atrophy; arthrogryposis; ophthalmic findings mainly focal pigmentary retina mottling, chorioretinal atrophy and/or coloboma, pallor, atrophy, increased excavation, hypoplasia and/or coloboma of optic nerve and abnormal visual function and low birthweight for gestational age” (line 312)

Line 401: how many studies reported on microcephaly?

Forty one 

Microcephaly was the most frequent sign found in neonates exposed to ZIKV during pregnancy among the articles in our review [25,35–38]. Forty one studies reported microcephaly [16,20,37,42,55–57,59–64,66,67,67,68,70–73,76–84,86–91,93–98]. (line 318)

The terms ZIKV, ZKV and Zika virus or Zika syndrome are all used. Please use the abbreviation specified in the introduction and keep this consistent throughout

The terms were corrected in the manuscript. 

Line 401-448: This entire section of the discussion just reiterates the results (as does most of the discussion). I would summarise and place into context- why is this important? What is new here?

In agreement with Reviewer 1, more thought it needed to describe what this review adds to the literature and what the limitations of the literature are.

What are the limitations of the review itself? You need a section describing the strengths and limitations, potential for bias in your results and issues with methodological rigour.

The manuscript would be improved with more careful thought in the discussion and presenting results in light of study limitations as well as discussion of the implications of the findings. At present, the discussion just describes the results again and there is a need for further synthesis/ summarising and description of what this study adds to what is already known.

The discussion has been rewritten. The changes will be described at the end of this letter.

Page 19 line 374 - "There were calcifications in five articles" change to "Calcifications were reported in five studies"

The text has been modified.

“Parenchymal or cerebellar calcification were described in 37 studies of our review [16,20,37,42,56,59–68,70–73,75–82,84,87,88,90,91,93–95,97,98] and also in other review which reported a prevalence of 42.6% [44]. These findings may also be found in congenital infections known by the mnemonic TORCH (Toxoplasmosis, Rubella, Cytomegalovirus and Herpes. However, the distribution of intracranial calcifications differs in each congenital infection: typically larger, denser and subcortical in congenital ZIKV, punctate and periventricular or cortical in CMV, diffuse and widely distributed in congenital toxoplasmosis and at basal ganglia in rubella [38,106–108]. Besides that it should be noted that some neurological changes (calcifications and cerebral atrophy) in congenital Zika syndrome are similar to other syndromes of infectious and also genetic etiology, such as Aicardi-Goutières syndrome [109].” (line 325)

Page 22 lines 450-451 - Unformatted reference? (ROBINSON et al., 451 2018)

The reference was formatted …“The pathophysiology of ZIKV infection and the mechanism of the virus’ passage across the placental barrier are still under investigation [45]. (line 383)

Table 1 - In the caption, it should be "...characteristics of the study SAMPLE" not POPULATION it has been corrected 

Table 1 - Words in Portuguese are scattered throughout the table (examples, País, imunohistoquimica)

Table 1 - change title of "POPULATION" column to "SAMPLE SIZE"

Table 1 - Correct typo in the column "MOTHER'S", not "MATHER'S"

Table 1 - What is "null" in Carvalho et al (2016) row, column CEPHALIC PERIMETER?

Table 1 - In DIAGNOSTIC METHOD(ZIKV) column some legend symbols are rendering as white squares. I suggest changing to letters or numbers if possible.

Table 1 - In English, decimal separators are points (.) not commas (,). Change all instances throughout Table 1

It has been corrected throughout the document.

Figure 3 - Improve quality of this figure. The numeric axis is unreadable and the text in the quality assessment categories are truncated. Font color should be black, not grey.

Figure 3 has been reviewed and modified.

Table 3 - Instead of "Associated factors" it should be "outcomes". Again, change "population" to "SAMPLE SIZE"

Table 3 - Why "associated factors" are repeated in this table? Reformat to reduce visual polution

Table 3 - Why some studies have the wording case/control and others have exposed/unexposed? Standardize nomenclature.

Table 3 - Why studies in the bottom part of the table 3 have contingency tables of case/control/whatever groups and the upper part does not?

Table 3 - Why "Not calculated"? If you have the contingency tables (as it seems to be the case) you could calculate yourselves.

Table 3 has been reviewed and modified: the association measures were calculated for Brazil (2016); the spreadsheet data were presented according to the measure of association used in the analysis: odds ratio for the case-control study and relative risk for cohort studies. Zin, 2016 was excluded because the comparative groups were missing.

References – Words in Portuguese (excluding the titles) are scattered throughout citations It has been corrected throughout the document.

Reviewer 2

Well written systematic review on Congenital Zika Syndrome (CZS).

The introduction could be strengthened to expand on international travelers.

Zika as arboviral disease in pregnancy should be briefly compared to other arboviral diseases in pregnancy, and this paper could be referred to:

Dengue, Zika and chikungunya during pregnancy: pre- and post-travel advice and clinical management.

Vouga M, Chiu YC, Pomar L, de Meyer SV, Masmejan S, Genton B, Musso D, Baud D, Stojanov M. J Travel Med. 2019 Dec 23;26(8):taz077. doi: 10.1093/jtm/taz077. PMID: 31616923 Free PMC article. Review.

The authors would do well highlighting how Zika could still re-emerge post 2016 outbreak in Asia and Africa where seroprevalence, and hence herd immunity, is not so high as in the Americas:

Zika in Angola and India.

Hamer DH, Chen LH. J Travel Med. 2019 Jun 11;26(5):taz012. doi: 10.1093/jtm/taz012.

Incidence of laboratory-confirmed Zika in Israeli travelers to Thailand: 2016-2019.

Leshem E, Lustig Y, Brosh-Nissimov T, Paran Y, Schwartz E. J Travel Med. 2019 Oct 14;26(7):taz057. doi: 10.1093/jtm/taz057.

We included in the discussion, line 307

“The ZIKV reemergence in regions with autochthonous transmission and the introduction of the virus in areas with established vector mosquito infestation may increase the risk of the development of congenital Zika syndrome in all regions of the world, especially in Africa, the Americas, Southeast Asia and the Western Pacific [158].

 In the absence of effective vaccines, licensed to date [164,165], and considering the complexity of effective vector control, innovative intersectoral strategies that transcend exclusive vector chemical control actions should be incorporated in areas with viral circulation [159–164]. Prevention strategies such as the use of repellents by pregnant women and condoms by sexual partners, both for travelers to regions with ZIKV circulation, and for residents of risk areas, as well as the implementation of strict screening protocols in the donation system of blood should be implemented. In addition, pregnant women and those intending to become pregnant may be advised to avoid unnecessary travel to endemic regions [165].” (line 468)

The introduction should also briefly highlight that Guillain-Barre Syndrome is the other feared complication of Zika.

It was described in the introduction:

“Most individuals infected with the Zika virus either do not develop symptoms or have mild and self-limited signs [12–14]. However, the disease has been linked to several neurologic manifestations in children and adults such as Guillain-Barré syndrome and peripheral nerve involvement, and ophthalmic complications such as retinal and optic nerve abnormalities.” (line 66)

The methods section should highlight the need for a good case definition:

Zika virus infection in pregnancy: Establishing a case definition for clinical research on pregnant women with rash in an active transmission setting.

Ximenes RAA, Miranda-Filho DB, Brickley EB, Montarroyos UR, Martelli CMT, Araújo TVB, Rodrigues LC, de Albuquerque MFPM, de Souza WV, Castanha PMDS, França RFO, Dhália R, Marques ETA; Microcephaly Epidemic Research Group (MERG). PLoS Negl Trop Dis. 2019 Oct 7;13(10):e0007763. doi: 10.1371/journal.pntd.0007763. eCollection 2019 Oct. PMID: 31589611 Free PMC article.

Not sure whether I overlooked it, but the issue of asymptomatic Zika infections in pregnancy and their fetal outcomes needs to be highlighted.

The need for a good case definition is highlighted in the introduction section: 

“...The laboratory diagnosis of ZIKV infection is limited by the high cost and cross-reaction with other flaviviruses [161,162], and then protocols for clinical diagnosis, in the context of simultaneous infection by other arboviruses, need to be implemented to define cases of ZIKV infection among pregnant women who have a rash” (lines 62-65). 

The issue of asymptomatic Zika infections is described in the discussion:

…“Congenital Zika syndrome is a serious public health problem, both because of the clinical severity of the cases and the extent of functional impairment. Absence of signs and symptoms at birth in exposed infants does not rule out their appearance later in childhood, thus highlighting the importance of structuring healthcare networks for comprehensive monitoring and care of these children. The high prevalence of asymptomatic cases at birth (65 to 83%) [5,157] can delay the identification of the association between congenital syndromes (with or without late onset) and ZIKV infection in the mother during pregnancy. Efforts are needed to provide care and support for all the needs of children with congenital Zika syndrome and their families, as well as effective organization of healthcare and social services.” (lines 457-466)

The discussion should end with the need for bringing a vaccine to the market and underpin the hurdles and challenges to indeed achieve this.

It is described in the discussion

…“In the absence of effective vaccines, licensed to date [164,165], and considering the complexity of effective vector control, innovative intersectoral strategies that transcend exclusive vector chemical control actions should be incorporated in areas with viral circulation [159–164]. Prevention strategies such as the use of repellents by pregnant women and condoms by sexual partners, both for travelers to regions with ZIKV circulation, and for residents of risk areas, as well as the implementation of strict screening protocols in the donation system of blood should be implemented. In addition, pregnant women and those intending to become pregnant may be advised to avoid unnecessary travel to endemic regions [165].” (line 471)

Other changes made to the text:

Abstract

We included the word usually in line 28

The signs and symptoms of Zika virus infection are usually mild and self-limited.

• We deleted ophthalmic and in line 29

• We included and; and also to in line 30

• We deleted prevalence of in line 32

• We deleted the word preestablished in line 35

We modified “..microcephaly, abnormal brain findings detected by imaging studies, significant changes in head circumference, low birthweight, low weight for gestational age, adverse pregnancy outcomes, central nervous system anomalies, corpus callosal anomalies, cerebral hyperechogenicity, and intrauterine fetal death.” lines 40-43. To: “...parenchymal or cerebellar calcifications, ventriculomegaly, CNS hypoplasia or atrophy, arthrogryposis, ocular findings in the posterior and anterior segments, abnormal visual function and low birthweight for gestational age.” (lines 40-42).

We modified “…It is thus necessary to develop effective preventive measures such as the use of specific repellents for pregnant women in areas of viral circulation and the establishment of effective public vector control measures” in lines 47-49.

To Our findings outline the disease profile in newborns and infants and may contribute to the development and updating of more specific clinical protocols. (lines 46-47).

Introduction

We replaced “…Zika virus (ZIKV) is a flavivirus of the family Flaviviridae, isolated in 1947, initially in non-human primates in Uganda, Africa, in humans in 1954 in Nigeria, in Aedes aegypti mosquitoes in 1969 in Malaysia, and in humans in 1977 in Indonesia [1–4]” in lines 52-54.

With “...Zika virus (ZIKV) is a flavivirus of the family Flaviviridae, isolated initially in non-human primates in Uganda (1947), and in humans (1954) in Nigeria, Africa[1–4] in lines 52-53.

We replaced “…introduction of ZIKV in Brazil, in 2013 or 2014 [7], the first cases of the disease were only reported in May 2015 [8]” in lines 59-60.

With “...In Brazil, the first cases of the disease were reported in May 2015 [6]” in lines 55-56.

• We replaced the word “perinatal” in line 64 with “vertical” in line 60

We included in line 62 The laboratory diagnosis of ZIKV infection is limited by the high cost and cross-reaction with other flaviviruses [12,13], and then protocols for clinical diagnosis, in the context of simultaneous infection by other arboviruses, need to be implemented to define cases of ZIKV infection among pregnant women who have a rash [14].

We replaced “…However, the disease has been linked to ophthalmic and neurological complications such as Guillain-Barré syndrome and peripheral nerve involvement” in lines 67-69.

With “...However, the disease has been linked to several neurologic manifestations in children and adults such as Guillain-Barré syndrome and peripheral nerve involvement, and ophthalmic complications such as retinal and optic nerve abnormalities” in lines 67-70.

• We replaced “besides various” in line 70 with “and also with” (line 71)

• We deleted lines 71-75

In Brazil, from 2015 to 2017, there were 433 suspected cases of ZIKV in fetuses involving miscarriages and stillbirths [30]. In 2018, 3,332 cases were reported in children with impaired growth and development. Among fetal, neonatal, or infant deaths related to ZIKV and other infectious etiologies, 357 cases were reported [31].

• We deleted lines 81-83

“…Because of the increase in reported cases of microcephaly and the appearance of other signs and symptoms in fetuses and newborns exposed to the Zika virus during pregnancy…”

• We replaced the word appearance in line 79 with onset (line 75)

• We replaced the word from in line 84 with in (line 79)

• We deleted”...until 84 November of the same year...” in line 84-85.

We replaced “...The detection occurred by clinical examination and imaging tests, which also reported brain calcifications. Children born to women infected with ZIKV between gestational weeks 16 and 18 presented more severe microcephaly (CDC, 2016). A cohort study of pregnant women conducted in Brazil described spontaneous abortions and other neurological changes related to ZIKV infection. There were four infants with microcephaly, 49 children had alterations in the clinical examination, in the imaging tests, or both, and 31 (63%) had abnormal neurological tests. The authors concluded that ZIKV infection during pregnancy is harmful to the fetus and is associated with fetal death, intrauterine growth restriction, and a spectrum of central nervous system abnormalities such as brain calcifications, atrophy, ventricular enlargement, and hypoplasia of central nervous system (CNS) structures [13,17,35] in lines 90-100.

With “...Detection of ZIKA infection during pregnancy has been found to be harmful to the fetus and can lead to fetal death and other abnormalities in newborns [12,16,34] in lines 85-87.

• We deleted lines 101-106

“Other clinical findings have been described in the literature, including craniofacial disproportion, brainstem dysfunction, seizures, irritability, limb contractures, spasticity, auditory and ocular abnormalities, and dysphagia, in addition to various abnormal neuroimaging findings [14,21,29,36–44]. These findings can also occur in similar congenital syndromes associated with other infectious diseases, such as toxoplasmosis, syphilis, varicella, parvovirus B1, rubella, cytomegalovirus, and herpes simplex [41,45–47].”

• We deleted lines 112-115

“Importantly, most of the published studies have been case reports and case series. They are usually based on early assessments which may help to describe the clinical picture, establish the natural history, and depict potential association of the virus with congenital syndromes.”

We replaced “...The characterization of congenital Zika syndrome is still challenging, mainly due to its recent discovery and the lack of robust studies with scientific evidence. This review aimed to determine the prevalence of signs and symptoms and characterize the congenital Zika syndrome based on a systematic review of the scientific literature” in lines 120-123. 

With “...This review aimed to determine the signs and symptoms that characterize the congenital Zika syndrome and contribute to a more accurate and timely diagnosis” in lines 95-97.

Methods

We repleced “...This review was performed with a preestablished protocol and described according to the recommendations of the Preferred Reporting Items for Systematic Reviews and Meta-Analyses statement [59]” in lines 129-131

With “…This review was performed with a protocol and described according to the recommendations of the Preferred Reporting Items for Systematic Reviews and Meta-Analyses statement [50] and is registrered in PROSPERO (CRD42020151754) in 27 October 2019. We used PICO (population, intervention, comparison, outcome) as a search strategy tool as described in S1 Table. in lines 104-108.

• We Included lines 114-116

“…Additionally, manual search was performed for bibliographic references of the selected articles and grey literature databases were also included to minimize publication bias [48–52].”

• We deleted lines 141-147 

Only 4 studies included in our review presented measures of association, therefore, it was not possible to perform a meta-analysis and to assess publication bias using statistical procedures. However, to minimize publication bias, we used a comprehensive search strategy [60,61]. 

Congenital Zika syndrome is a disease that is still being characterized, so data from observational studies with a limited number of participants and no measure of association were included.

• We fixed the typos in line 153 (DAF and WB) (line 125)

• We deleted lines 159-161

“... entomological studies, molecular biology research, and in vitro studies; social and psychological research, studies in public policy; descriptive studies on epidemiological profile, research on vaccines and prevention”.

We replaced “...Data were extracted by one of the authors (DFA) and reviewed independently by the other (MDW)” in lines 167-168

With “…Data were extracted by two authors independently (DAF and WB), and reviewed by the other (MDW)” in lines 135-136.

• We replaced the word elaborated in line 169 with made (line 137)

• We included the word observational in line 175 (line 143)

• We replaced “…To assess the quality of case reports and case series, the Grading of Recommendations, Evaluation, Development, and Evaluation (GRADE) was used. The original instrument consists of 8 items, but we used 5 items, because the others are more relevant for studies on adverse drug events, which was not the purpose of our review [62] in lines 177-181. 

• With “...To assess the quality of case reports, we used JBI (Joanna Briggs Institute) critical appraisal checklist for case reports [53]” in lines 145-146.

• We replaced the word LG ( line 181) with DAF (line 147)

• We replaced “…A description was performed of the studies regarding...” line 204

• With “...The studies were described according to the country in line 151.

• We deleted line 207-209

Odds ratios (OR) and respective confidence intervals (CI) were presented for studies that performed multivariate analysis of factors associated with congenital Zika virus infection [63].

• We Included the word findings in line 152.

• We replaced “…The quality of articles was assessed with the MINORS instrument for all observational studies and with the Grading of Recommendations, Assessment, Development, and Evaluation (GRADE) approach for case series and case reports. Comparative studies (n=3) achieved good quality criteria, while most did not report adequately on the following criteria: loss to follow-up less than 5% and prospective calculation of the study size (Fig 2). At least 93% of the care reports and case series showed excellent quality standards, but 26% had not performed sufficiently long follow-up for outcomes to occur (Fig 3). Most of the studies were performed in Brazil (59%), the epicenter of reported cases (Table 1)” in lines 215-223.

• With “…Among the articles, more than 90% presented end points appropriate to their aim and consecutive inclusion of patients according to Minors. Less than 20% of the studies reported adequately on the following criteria: “loss to follow-up less than 5%”, “prospective calculation of the study size”, “an adequate control group”, “Contemporary groups”, “baseline equivalence of groups” and “adequate statistical analyses” (Fig 2). From all case reports 36 % did not describe patient´s history adequately. The other items were adequate in more than 90% of studies and two were not applicable (Fig 3)

…Most of the studies were performed in Brazil (59%), the epicenter of reported cases. Regarding study design, most of the studies were case reports (n=22) or case series (n=20), and there were two cohort studies, one case-control, and one cross-sectional study. All the studies reported at least one laboratory method for the diagnosis of women and/or children exposed to Zika virus during pregnancy. Most of the studies described signs and symptoms in children exposed to Zika virus in the first and second trimesters (Table 1)” lines 163-169 and 178-184.

• We replaced figures 2 and 3. 

• We replaced Tables 1 and 3 

• We included references in lines 215-217, 219-222, 227-231, 242-243, 245-246, 267-276, 278-285, 289-292, 300-302

• We deleted lines 263-264

 Neurological signs and symptoms were the ones most frequently described in both the imaging tests and autopsies 

• We replaced “…Four Four articles described the association between congenital ZIKV infection, microcephaly, and other adverse pregnancy outcomes. One case-control study demonstrated the association between microcephaly and ZIKV, tested by Zika virus-specific IgM and quantitative RT-PCR in serum and cerebrospinal fluid of neonates with microcephaly and two controls, while maternal serum samples were tested by plaque reduction neutralization assay for Zika virus and dengue virus. In addition to microcephaly, De Araújo et al. (2016) [21] detected congenital malformations through abnormal brain findings on imaging tests in children with virus in the serum or cerebrospinal fluid and serum only, and anthropometric alterations such as significant changes in head circumference, birth weight, and weight for gestational age. Brasil et al. (2017) described adverse pregnancy outcomes, including cases of fetal loss in every trimester and the need for emergency cesarean sections [17]. In a case series of children exposed to ZIKV, an association was found between ocular changes and first-trimester infection, microcephaly, arthrogryposis, and other central nervous system (CNS) lesions [65] (Table 3). ZIKV infection was associated with CNS anomalies, corpus callosum anomalies, cerebral hyperechogenicity, and intrauterine fetal death [41]” in lines 266-282.

• With “...Three articles described the association between congenital ZIKV infection, microcephaly, and other adverse pregnancy outcomes [16,20,55]. One case-control study demonstrated the association between microcephaly and ZIKV in neonates tested by Zika virus-specific IgM and quantitative RT-PCR in serum and cerebrospinal fluid. In addition to microcephaly, this study detected congenital malformations through abnormal brain findings on imaging tests in children with virus in the serum or cerebrospinal fluid and serum only [20]. One cohort study described adverse pregnancy outcomes, including cases of fetal loss in every trimester [16]. Another cohort study identified an association between ZIKV infection and CNS anomalies, cerebral hyperechogenicity, and intrauterine fetal death [55] (Table 3)” in lines 247-256.

• We replaced “…A frequently observed clinical sign was arthrogryposis, reported in 12 articles. Other osteoskeletal signs included clubfoot and hip dysplasia. Morphological changes of the head were described in ten articles and overriding sutures or closed fontanels in five. 

• The most frequent ophthalmic changes were in the posterior segment, found in 22% of the reviewed articles that presented abnormalities of the retina and optic nerve [29,65,67,73,73,74,88,91,96]. Abnormalities of visual function were described in 11% of the articles [67,73,84,86,88], as well as extrinsic ocular motility [65,67,74,84]. Abnormalities in the anterior segment appeared in 9% of the articles [74,91,98] and refractive error in 4% [67,91] (Table 2)” in lines 322-330.

• With “...A frequently observed clinical sign was arthrogryposis, reported in 12 articles [37,58,66,68,70,72,77,82–84,87,89]. Other osteoskeletal signs included clubfoot [82,84] and hip dysplasia [16,68]. Morphological changes of the head were described in ten articles [16,37,62,66,68,70,77,80,81,84] and overriding sutures or closed fontanels in five [67,68,77,80,81]. 

The most frequent ophthalmic abnormalities occurred in the posterior segment, found in 22% of the reviewed articles which included lesions of the retina ( focal pigmentary retina mottling, chorioretinal atrophy and/or coloboma) and optic nerve (pallor, atrophy, increased excavation, hypoplasia and/or coloboma) [28,57,59,66,66,67,81,84,89]. Abnormalities of visual function were described in 11% of the articles [59,66,77,79,81], as well as extrinsic ocular motility (nystagmus and strabismus)[57,59,67,77]. Abnormalities in the anterior segment (cataract and glaucoma) appeared in 9% of the articles [67,84,91] and refractive error (myopia, hyperopia and astigmatism) in 4% [59,84]. Lines 218-231

• Lesions in other organ systems were also observed, to a lesser extent: cardiovascular system, [16], genitourinary system (ambiguous genitalia [72] and bilateral cryptorchidism [37,81]), and gastrointestinal system (dysphagia [16,68]) (Table 2). Other clinical features included small for gestational age (SGA) in six studies [16,20,66,81,82,87], hearing abnormalities in four [16,56,60,84], and unilateral diaphragmatic paralysis in three articles [37,67,70] (Table 2)” in lines 241-246.

• We replaced titles: 

Table 2: Clinical signs of children exposed to ZIKV in pregnancy. 

Table 4: Signs detected on imaging tests of children exposed to Zika virus (ZIKV) in pregnancy. 

Table 5: Autopsy findings of children exposed to Zika virus (ZIKV) in pregnancy. 

• We deleted in line 332: (congenital heart disease [n=1])

• We deleted in line 333: [n=1], [n=2] and in line 334 [n=2].

• We deleted in line 342: increased fluid spaces 

• We replaced “…The following other lesions were observed on imaging tests in a cohort study” in line 369

• With The following other lesions were observed on imaging tests in one study [16]” in line 285.

We deleted line 375-376 “…in three and two studies, respectively.”

Discussion

We replaced “...In our review, the main signs and symptoms associated with congenital Zika virus syndrome were microcephaly, abnormal brain findings detected by imaging studies (calcifications, CNS hypoplasia, ventriculomegaly, corpus callosum anomaly, cerebral hyperechogenicity), low birthweight for gestational age, adverse pregnancy outcomes (miscarriage, emergency cesarean section, fetal loss), and intrauterine fetal death in lines 396-400.

With “...In our review, the main signs and symptoms associated with congenital Zika virus syndrome were microcephaly, parenchymal or cerebellar calcifications, ventriculomegaly, CNS hypoplasia or atrophy; arthrogryposis; ophthalmic findings mainly focal pigmentary retina mottling, chorioretinal atrophy and/or coloboma, pallor, atrophy, increased excavation, hypoplasia and/or coloboma of optic nerve and abnormal visual function and low birthweight for gestational age” in lines 313-317.

We deleted the word also in line 413

We deleted words likewise and other in line 414

We replaced “...Parenchymal or cerebellar calcification was a common sign described by the studies in the review, similar to the characteristics found in congenital cytomegalovirus (CMV) infection. However, the distribution of intracranial calcifications differs, typically subcortical in congenital ZIKV and periventricular in CMV [51,117,118]. It should be noted that the neurological changes (calcifications and cerebral atrophy) in congenital Zika syndrome show similarities to other syndromes of infectious and also genetic etiology, such as Aicardi-Goutières syndrome [119] in lines 423-429.

• With “...Parenchymal or cerebellar calcification were described in 37 studies of our review [16,20,37,42,56,59–68,70–73,75–82,84,87,88,90,91,93–95,97,98] and also in other review which reported a prevalence of 42.6% [44]. These findings may also be found in congenital infections known by the mnemonic TORCH (Toxoplasmosis, Rubella, Cytomegalovirus and Herpes. However, the distribution of intracranial calcifications differs in each congenital infection: typically larger, denser and subcortical in congenital ZIKV, punctate and periventricular or cortical in CMV, diffuse and widely distributed in congenital toxoplasmosis and at basal ganglia in rubella [38,106–108]. Besides that it should be noted that some neurological changes (calcifications and cerebral atrophy) in congenital Zika syndrome are similar to other syndromes of infectious and also genetic etiology, such as Aicardi-Goutières syndrome [109] lines 325-335.

• We included lines 353-356

In short, these findings can also occur in similar congenital syndromes associated with other infectious diseases, such as toxoplasmosis, syphilis, varicella, parvovirus B1, rubella, cytomegalovirus, and herpes simplex [20,39,55,115] but the signs and symptoms presented by CZS seem to be more serious. 

• We replaced “...The osteoskeletal system was the second most frequently altered organ system in newborns exposed in utero to ZIKV, featuring arthrogryposis [13,17,66,68,83,89–431 91,113]” in lines 430-432.

• With “...The osteoskeletal system was the second most frequently altered organ system in newborns exposed in uterus to ZIKV and arthrogryposis was described in 18 studies reviewed [16,37,58,66–68,70–72,77,78,82–84,87,89,94,116], as observed by other studies, mainly in more severe cases [12,44,76,110] lines 357-360.

• We replaced …“Ocular findings were described in 15 articles as anatomical changes, and five of these articles also reported functional changes. However, many of these anatomical changes may impact the child’s visual function in the future, since they affect prime areas of the eye. Although ocular alterations have been detected in children with microcephaly and other brain anomalies [65], Zin (2017) identified ophthalmic lesions in 7% of children who had no CNS abnormalities, suggesting that ocular abnormalities may be key findings in the syndrome. Importantly, since the child’s first contact with the environment is through eyesight, children with visual impairment commonly experience delayed neuropsychomotor development, which, alongside the syndrome’s other signs and symptoms, hinder the child’s ability to integrate with his or her surroundings [120]. Posterior segment ocular alterations have been described as possible features of the severe form of the syndrome [17], as described in other studies [121,122]. Additionally, Ventura identified posterior segment alterations in 44% of children and abnormal visual function in 100%, showing that cerebral cortical involvement is also an important cause of visual impairment in these children [67]. Alterations in the anterior segment were reported by 9% of the selected studies, also observed in others [123] “lines 433-448.

• With …“Anatomical [57,59,66–68,70,75,81,84,89] and functional [59,60,68,77,81] changes of the eye were described in 12 articles, such as posterior and anterior segment abnormalities, extrinsic eye motility and abnormal visual function. Many of these anatomical changes may impact the child’s visual function in the future, since they affect prime areas of the eye. Importantly, since the child’s first contact with the environment is through eyesight, children with visual impairment commonly experience delayed neuropsychomotor development, which, alongside the syndrome’s other signs and symptoms, hinder the child’s ability to integrate with his or her surroundings [117]. These ocular findings were described also in other studies descripted [44,118–120]” in lines 361-369.

• We included in lines 370-382

“TORCH infections have also been associated with serious ophthalmic alterations. In congenital toxoplasmosis infection, chorioretinal lesions are usually bilateral and can present with active lesions or regressed scar [57,121]. Congenital cytomegalovirus and herpes virus infections can also manifest with active ocular inflammation. In Zika virus infection, however, there has been no active ocular inflammation cases reported so far [121]. There is usually not pigmentary mottling seen outside of areas of chorioretinal atrophy in congenital toxoplasmosis infection, as seen in zika virus infection [57]. In congenital rubella infection, the pigment mottling is usually diffused compared with the focal pigment mottling seen in zika virus infection [57]. Congenital cytomegalovirus can present with chorioretinal lesions similar to congenital toxoplasmosis but less heavily pigmentated and with pale or small optic disc [122,123]. Optic nerve hypoplasia, commonly seen in ZIKV congenital infection, is rarely seen in rubella, toxoplasmosis and herpes congenital infections [57].”

• We included lines 444-445

“Recent studies have identified neurological disorders such as delay in neurodevelopment, mainly in the domain of language in children exposed to ZIKV who were asymptomatic at birth. [124,146].”

We excluded (ROBINSON et al., 2018) in lines 450-451

We replaced “Eight studies in our review reported placental alterations [41,50,72,85,88,94,97,106]” in line 453.

With “Placental alterations were reported in eight studies [37,55,65,78,81,87,90,100]” in line 387.

We included the following sentencs in lines 406-413:

 “…possibly due to the limited access to those exams in some settings. Obstetric ultrasound proved to be useful for tracking ZIKV-related brain injury in prenatal care in pregnant women exposed to ZIKV. Although less accurate than CT and MRI it is available in many health facilities [141,142].”

“…One point to highlight in our review is the anatomopathological analysis. It was possible to notice either the severity of the systemic impairment of the cases that died, especially those whose maternal infection occurred in the first trimester of pregnancy and the correspondence of clinical and anatomopathological findings.”

We replaced …”Several studies reported structural changes in the genitourinary system [41,50,79,143]. Villamil-Gómez (2019) detected severe bilateral renal hypoplasia in fetal autopsies [144]. Recent studies in older children have detected bladder impairment and possible kidney damage such as very low bladder capacity, bladder hyperactivity with increased consistency, high bladder pressure during the filling phase, and high postvoid residual volume (PVR) and/or recurrent urinary tract infection [145]. Congenital cardiovascular anomalies were identified in children exposed to ZIKV [41,68,104,130]. Other studies have also described cardiological symptoms such as complex congenital heart disease, echocardiographic abnormalities, and cardiac overload [146–149]” in lines 490-498.

With “...Structural changes in the genitourinary system were reported [40,58,75,148] and were also described by Villamil-Gómez (2019) in fetal autopsies [149]. Bladder impairment and possible kidney damage such as very low bladder capacity, bladder hyperactivity with increased consistency, high bladder pressure during the filling phase, and high postvoid residual volume (PVR) and/or recurrent urinary tract infection were reported in older children [150]. Congenital cardiovascular anomalies were identified in children exposed to ZIKV such as cardiomyopathy, hyperechogenicity of aortic valve, mitral valve, and aortic root [19,58,64,100]. Other studies have also described cardiological symptoms and imaging findings such as complex congenital heart disease, echocardiographic abnormalities, and cardiac overload [151–154]. Lines 422-432

We replaced “Recent studies have identified neurological disorders in children exposed to ZIKV who were asymptomatic at birth [137,151]. Such findings are relevant, since there is evidence that ZIKV can continue to replicate in the infant’s brain after birth [152], and that cerebral growth of infants exposed during pregnancy can decelerate, even after birth [73]”. lines 511-515.

With “…Recent studies have identified neurological disorders such as delay in neurodevelopment, mainly in the domain of language in children exposed to ZIKV who were asymptomatic at birth. [131,153]. Such findings are relevant, since there is evidence that ZIKV can continue to replicate in the infant’s brain after birth [154], and that cerebral growth of infants exposed during pregnancy can decelerate, even after birth [66] “in lines 444-448.

• We included the following sentences in lines 468-480

…“The ZIKV reemergence in regions with autochthonous transmission and the introduction of the virus in areas with established vector mosquito infestation may increase the risk of the development of congenital Zika syndrome in all regions of the world, especially in Africa, the Americas, Southeast Asia and the Western Pacific [158]. “ …In the absence of effective vaccines, licensed to date [164,165], and considering the complexity of effective vector control, innovative intersectoral strategies that transcend exclusive vector chemical control actions should be incorporated in areas with viral circulation [159–164]. Prevention strategies such as the use of repellents by pregnant women and condoms by sexual partners, both for travelers to regions with ZIKV circulation, and for residents of risk areas, as well as the implementation of strict screening protocols in the donation system of blood should be implemented. In addition, pregnant women and those intending to become pregnant may be advised to avoid unnecessary travel to endemic regions [165]”. 

• We deleted lines 539-541

We replaced” …One limitation to the study was the lower number of observational clinical studies. The inclusion of case series and case reports, with lower levels of scientific evidence, is justified for the investigation of new and rare diseases such as congenital Zika syndrome, since evidence based on clinical case reports is necessary to understand the illness process [157] in lines 535-539

With “…One limitation to the study was the low number of comparative clinical studies. The inclusion of case series and case reports, with lower levels of scientific evidence, is justified for the investigation of new and rare diseases such as congenital Zika syndrome, since evidence based on clinical case reports is necessary to understand the natural history of a new illness [166]” in lines 481-485.

---

## [Decision Letter · Decision Letter 1]

2 Nov 2020

Congenital Zika Syndrome: a systematic review

PONE-D-20-18691R1

Dear Dr. Freitas,

We’re pleased to inform you that your manuscript has been judged scientifically suitable for publication and will be formally accepted for publication once it meets all outstanding technical requirements.

Kind regards,

Kazumichi Fujioka

Academic Editor

PLOS ONE

Additional Editor Comments (optional):

Reviewers' comments:

Reviewer's Responses to Questions

**Comments to the Author**

1. If the authors have adequately addressed your comments raised in a previous round of review and you feel that this manuscript is now acceptable for publication, you may indicate that here to bypass the “Comments to the Author” section, enter your conflict of interest statement in the “Confidential to Editor” section, and submit your "Accept" recommendation.

Reviewer #1: All comments have been addressed

Reviewer #2: All comments have been addressed

2. Is the manuscript technically sound, and do the data support the conclusions?

Reviewer #1: Yes

Reviewer #2: Yes

3. Has the statistical analysis been performed appropriately and rigorously? 

Reviewer #1: Yes

Reviewer #2: I Don't Know

4. Have the authors made all data underlying the findings in their manuscript fully available?

Reviewer #1: Yes

Reviewer #2: Yes

5. Is the manuscript presented in an intelligible fashion and written in standard English?

Reviewer #1: Yes

Reviewer #2: Yes

6. Review Comments to the Author

Reviewer #1: The authors re-estructured several parts of the manuscript, improving the overall presentation of the work, its conclusions and limitations.

Minor reviews

The supplementary material tables are out of order, and the PRISMA Checklist seems to be repeated, tooking the place of the PICO table. Check thoroughly all material and their order.

Supplementary Table S5: score ranges are absent and the JBI table contains words in Portuguese.

The authors should have more care when organizing the data and actually review the manuscript PDF proof to ensure that all information is present and in the proper order. Most critiques given in this review could have be avoided if the authors performed a more careful review of all material before submission.

Reviewer #2: All comments were adequately addressed. All comments were adequately addressed. All comments were adequately addressed. All comments were adequately addressed. All comments were adequately addressed. All comments were adequately addressed. All comments were adequately addressed. All comments were adequately addressed. All comments were adequately addressed.

7. PLOS authors have the option to publish the peer review history of their article (what does this mean?). If published, this will include your full peer review and any attached files.

Reviewer #1: No

Reviewer #2: No

---

## [Editor Report · Acceptance letter]

16 Nov 2020

PONE-D-20-18691R1 

Congenital Zika Syndrome: a systematic review 

Dear Dr. Freitas:

I'm pleased to inform you that your manuscript has been deemed suitable for publication in PLOS ONE. Congratulations! Your manuscript is now with our production department. 

Kind regards, 

on behalf of

Dr. Kazumichi Fujioka 

Academic Editor

PLOS ONE